# Decentralized Stochastic Gradient Descent Ascent for Finite-Sum Minimax Problems

## Abstract

Minimax optimization problems have attracted significant attention in recent years due to their widespread application in numerous machine learning models. To solve the minimax problem, a wide variety of stochastic optimization methods have been proposed. However, most of them ignore the distributed setting where the training data is distributed on multiple workers. In this paper, we developed a novel decentralized stochastic gradient descent ascent method for the finite-sum minimax problem. In particular, by employing the variance-reduced gradient, our method can achieve $O(\frac{\sqrt{n}\kappa^3}{(1-\lambda)^2\epsilon^2})$ sample complexity and $O(\frac{\kappa^3}{(1-\lambda)^2\epsilon^2})$ communication complexity for the nonconvex-strongly-concave minimax problem. As far as we know, our work is the first one to achieve such theoretical complexities for this kind of minimax problem. At last, we apply our method to optimize the AUC maximization problem, and the experimental results confirm the effectiveness of our method.

## 1 Introduction

In this paper, we consider the following decentralized finite-sum minimax problem:

$$\min_{\mathbf{x}\in\mathbb{R}^d}\max_{\mathbf{y}\in\mathbb{R}^{d'}} F(\mathbf{x},\mathbf{y}) \triangleq \frac{1}{K}\sum_{k=1}^{K}\left(\frac{1}{n}\sum_{i=1}^{n}f_i^{(k)}(\mathbf{x},\mathbf{y})\right) . \tag{1}$$

It is assumed that there are totally $K$ workers in a decentralized training system. Each worker has its own dataset and objective function $f^{(k)}(\mathbf{x},\mathbf{y}) = \frac{1}{n}\sum_{i=1}^{n} f_i^{(k)}(\mathbf{x},\mathbf{y})$ where $f_i^{(k)}(\mathbf{x},\mathbf{y})$ is the loss function for the $i$-th sample on the $k$-th worker and $n$ is the total number of samples on each worker. In this paper, $f_i^{(k)}(\mathbf{x},\mathbf{y})$ is assumed to be nonconvex in $\mathbf{x}$ and $\mu$-strongly-concave in $\mathbf{y}$. Under this kind of decentralized setting, all workers collaboratively optimize Eq. (1) to learn the model parameter $\mathbf{x}$ and $\mathbf{y}$.

The minimax optimization problem in Eq. (1) covers numerous machine learning models, such as adversarial training Goodfellow et al. (2014a;b); Madry et al. (2017), distributionally robust optimization Lin et al. (2020); Luo et al. (2020), AUC maximization Ying et al. (2016); Liu et al. (2019a), etc. Recently, many efforts have been devoted to developing efficient optimization algorithms to solve the minimax optimization problem. For instance, Lin et al. (2020) proposed a stochastic gradient descent ascent method and investigated its convergence rate. Afterwards, several accelerated methods Luo et al. (2020); Xu et al. (2020); Qiu et al. (2020) have been proposed to improve the convergence speed by utilizing the variance reduction technique or momentum strategy. However, these methods only focus on the single-machine setting. It's unclear how these methods converge under the decentralized setting and how large their communication complexities are.

To handle the large-scale minimax optimization problem, some distributed methods have been proposed in recent years. In Deng & Mahdavi (2021), a communication-efficient local stochastic gradient descent ascent method was proposed, whose convergence rate was further improved in Xie et al. (2021) by resorting to the variance reduction technique. Guo et al. (2020) deveoped CoDA for the AUC maximization problem. However, these methods are based on the parameter-server setting so that they are not applicable to our decentralized setting. Recently, Liu et al. (2019b) developed a decentralized optimistic stochastic gradient

method and established the convergence rate for the nonconvex-nonconcave problem. Xian et al. (2021) developed a decentralized stochastic variance-reduced gradient descent ascent method for the nonconvex-strongly-concave problem based on the STORM gradient estimator Cutkosky & Orabona (2019). However, it has a large communication complexity $O(1/\epsilon^3)$ to achieve the $\epsilon$-accuracy solution [1]. On the contrary, the decentralized algorithm for minimization problems can achieve the $O(1/\epsilon^2)$ communication complexity. Moreover, Xian et al. (2021) only studied the stochastic setting, failing to handle the finite-sum optimization problem. Recently, Zhang et al. (2021b) reformulated the policy evaluation problem in reinforcement learning as a finite-sum minimax problem and then proposed the decentralized GT-SRVR method to solve it. However, this method requires to compute the full gradient periodically, incurring large computation overhead.

To overcome aforementioned issues, we developed a novel decentralized stochastic gradient descent ascent (DSGDA) method for optimizing Eq. (1) efficiently. In detail, on each worker, DSGDA computes the variance-reduced gradient based on the local dataset and then employs the gradient tracking communication scheme to update the local model parameters $\mathbf{x}$ and $\mathbf{y}$. Furthermore, we established the convergence rate of DSGDA for the finite-sum nonconvex-strongly-concave problem. Specifically, our theoretical analysis shows that DSGDA can achieve $O(\frac{\kappa^3}{(1-\lambda)^2\epsilon^2})$ communication complexity, which is better than $O(\frac{\kappa^3}{(1-\lambda)^2\epsilon^3})$ of Xian et al. (2021) and matches $O(\frac{\kappa^3}{(1-\lambda)^2\epsilon^2})$ of Zhang et al. (2021b) in terms of the order of the solution accuracy $\epsilon$, where $1-\lambda$ represents the spectral gap of the communication network and $\kappa$ denotes the condition number of the loss function. Moreover, our method can achieve $O(\frac{\sqrt{n}\kappa^3}{(1-\lambda)^2\epsilon^2})$ sample complexity on each worker, which is better than $O(n + \frac{\sqrt{n}\kappa^3}{(1-\lambda)^2\epsilon^2})$ of Zhang et al. (2021b)[2] in terms of $n$ because our method does not need to periodically compute the full gradient as Zhang et al. (2021b). To the best of our knowledge, there is no existing literature achieving such a favorable sample complexity. This confirms the superiority of our method. The detailed comparison between our method and existing methods is demonstrated in Table 1. At last, we apply our method to optimize the decentralized AUC maximization problem and the experimental results confirm the superior empirical performance of our method. Finally, we summarize the contribution of our work in the following.

- We developed a novel decentralized stochastic gradient descent ascent method for optimizing finite-sum nonconvex-strongly-concave minimization problems without periodically computing the full gradient as existing methods Luo et al. (2020); Zhang et al. (2021b). Therefore, our method is efficient in computation.

- We established the convergence rate for our proposed decentralized optimization method, which demonstrates that our method can achieve a better sample complexity than existing decentralized minimax optimization methods. This is the first work achieving such a theoretical result for the decentralized minimax problem.

- We conducted extensive experiments on the AUC maximization problem, which confirms the effectiveness of our method in practical applications.

| Methods | Sample Complexity | Communication Complexity | Category |
|---|---|---|---|
| DM-HSGD Xian et al. (2021) | $O(\frac{\kappa^3}{(1-\lambda)^2\epsilon^3})$ | $O(\frac{\kappa^3}{(1-\lambda)^2\epsilon^3})$ | Stochastic |
| GT-SRVR Zhang et al. (2021b) | $O(n + \frac{\sqrt{n}\kappa^3}{(1-\lambda)^2\epsilon^2})$ | $O(\frac{\kappa^3}{(1-\lambda)^2\epsilon^2})$ | Finite-sum |
| Ours | $O(\frac{\sqrt{n}\kappa^3}{(1-\lambda)^2\epsilon^2})$ | $O(\frac{\kappa^3}{(1-\lambda)^2\epsilon^2})$ | Finite-sum |

Table 1: The comparison in sample and communication complexities between our method and baseline methods. Here, $\kappa$ is the condition number, $1-\lambda$ is the spectral gap, and $n$ is the number of samples on each worker.

---

[1]Here, we omit the spectral gap and condition number for simplification.
[2]The first term is ignored in Zhang et al. (2021b).

## 2 Related Work

### 2.1 Minimax Optimization

Minimax optimization has attracted a surge of attention in the machine learning community in the past few years due to its widespread application in many machine learning models. To this end, a line of research is to develop efficient optimization methods Sanjabi et al. (2018); Nouiehed et al. (2019); Jin et al. (2020); Yan et al. (2020); Zhang et al. (2021a); Chen et al. (2020); Yang et al. (2020); Tran-Dinh et al. (2020) to solve the minimax optimization problem. In particular, under the *stochastic setting*, Lin et al. (2020) developed a single-loop stochastic gradient descent ascent (SGDA) method, which updates $\mathbf{x}$ and $\mathbf{y}$ for only one step with stochastic gradients in each iteration. The sample complexity of SGDA for the nonconvex-strongly-concave minimax problem is $O(\kappa^3/\epsilon^4)$. Later, Qiu et al. (2020); Guo et al. (2021) combined the momentum technique with SGDA to accelerate the empirical convergence speed. Moreover, Qiu et al. (2020); Huang et al. (2020b) utilized the variance reduction technique STORM Cutkosky & Orabona (2019) to accelerate the convergence speed for nonconvex-strongly-concave minimax problems.

As for the *finite-sum setting*, Luo et al. (2020) proposed a double-loop (SREDA) method, which updates $\mathbf{x}$ for one step with the variance-reduced gradient estimator SPIDER Fang et al. (2018) and solves the maximization problem about $\mathbf{y}$ with multiple gradient ascent steps. As such, it can achieve the $O(n + n^{1/2}\kappa^2/\epsilon^2)$ sample complexity for the *finite-sum* nonconvex-strongly-concave minimax problem. However, SREDA requires to periodically compute the full gradient, which is not practical for large-scale real-world applications. In addition, its step size should be as small as $\epsilon$, which also limits its application for real-world tasks. Recently, Xu et al. (2020) resorted to the SpiderBoost Wang et al. (2019) variance reduction technique to tolerate a large step size. But it still needs to compute the full gradient periodically so that it has the same sample complexity with SREDA.

### 2.2 Decentralized Optimization

In recent years, decentralized optimization methods have been applied to optimize large-scale machine learning models. In particular, Lian et al. (2017) proposed a decentralized stochastic gradient descent (DSGD) method based on the gossip communication scheme, while Pu & Nedić (2020); Lu et al. (2019) used the gradient tracking communication scheme for DSGD. Yu et al. (2019) applied the momentum technique to DSGD to accelerate the convergence speed. Afterwards, the variance reduction technique has been utilized to further accelerate the convergence speed of DSGD. For example, Sun et al. (2020) combines SPIDER Fang et al. (2018) with the gradient-tracking-based DSGD, achieving the near-optimal sample and communication complexity. Besides, there are some works focusing on the communication-efficient methods by compressing gradients Koloskova et al. (2019) or skipping communication rounds Li et al. (2019). However, all these methods are designed for the minimization problem. Hence, they are not applicable to optimize Eq. (1).

A few efforts have been made to optimizing the decentralized minimax problem in the past two years. For example, Liu et al. (2019b) developed a decentralized optimistic stochastic gradient method to train the nonconvex-nonconcave generative adversarial nets Goodfellow et al. (2014a). Rogozin et al. (2021) focused on the strongly-convex-strongly-concave problem. Recently, Beznosikov et al. (2021) proposed a communication-efficient method based on the stochastic extragradient algorithm. Xian et al. (2021) developed a decentralized stochastic gradient descent ascent method based on the STORM Cutkosky & Orabona (2019) gradient estimator for the stochastic minimax problem, rather than the finite-sum problem. Zhang et al. (2021b) proposed GT-SRVR based on the SPIDER gradient estimator Zhang et al. (2021b) for finite-sum problems, which requires to periodically compute the full gradient.

## 3 Efficient Decentralized Stochastic Gradient Descent Ascent Method

### 3.1 Problem Setup

In this paper, the communication network in the decentralized training system is represented by $\mathcal{G} = \{P, W\}$. Here, $P = \{p_1, p_2, \cdots, p_K\}$ represents $K$ workers. $W = [w_{ij}] \in \mathbb{R}^{K \times K}$ is the adjacency matrix, denoting

---

**Algorithm 1** Efficient Decentralized Stochastic Gradient Descent Ascent (DSGDA)

---

**Require:** $\mathbf{x}_0^{(k)} = \mathbf{x}_{-1}^{(k)} = \mathbf{x}_0$, $\mathbf{y}_0^{(k)} = \mathbf{y}_{-1}^{(k)} = \mathbf{y}_0$, $\mathbf{v}_{-1}^{(k)} = \mathbf{a}_{-1}^{(k)} = 0$, $\mathbf{u}_{-1}^{(k)} = \mathbf{b}_{-1}^{(k)} = 0$,

1: $\mathbf{g}_{i,-1}^{(k)} = 0$, $\mathbf{h}_{i,-1}^{(k)} = 0$ for $i \in \{1, 2, \cdots n\}$.

2: **for** $t = 0, \cdots, T - 1$ **do**

3:     Randomly select samples $\mathcal{S}_t$ with $|\mathcal{S}_t| = s_t$ and then compute $\mathbf{v}_t^{(k)}$ and $\mathbf{u}_t^{(k)}$ as Eq. (2) and Eq. (4)

4:     Update $\mathbf{x}$:

$$\mathbf{a}_t^{(k)} = \sum_{j \in \mathcal{N}_k} w_{kj} \mathbf{a}_{t-1}^{(j)} + \mathbf{v}_t^{(k)} - \mathbf{v}_{t-1}^{(k)}$$

$$\mathbf{x}_{t+\frac{1}{2}}^{(k)} = \sum_{j \in \mathcal{N}_k} w_{kj} \mathbf{x}_t^{(j)} - \gamma_1 \mathbf{a}_t^{(k)}$$

$$\mathbf{x}_{t+1}^{(k)} = \mathbf{x}_t^{(k)} + \eta(\mathbf{x}_{t+\frac{1}{2}}^{(k)} - \mathbf{x}_t^{(k)})$$

5:     Update $\mathbf{y}$:

$$\mathbf{b}_t^{(k)} = \sum_{j \in \mathcal{N}_k} w_{kj} \mathbf{b}_{t-1}^{(j)} + \mathbf{u}_t^{(k)} - \mathbf{u}_{t-1}^{(k)}$$

$$\mathbf{y}_{t+\frac{1}{2}}^{(k)} = \sum_{j \in \mathcal{N}_k} w_{kj} \mathbf{y}_t^{(j)} + \gamma_2 \mathbf{b}_t^{(k)}$$

$$\mathbf{y}_{t+1}^{(k)} = \mathbf{y}_t^{(k)} + \eta(\mathbf{y}_{t+\frac{1}{2}}^{(k)} - \mathbf{y}_t^{(k)})$$

6:     Update $\mathbf{g}$ and $\mathbf{h}$:

$$\mathbf{g}_{i,t}^{(k)} = \begin{cases} \nabla_{\mathbf{x}} f_i(\mathbf{x}_t^{(k)}, \mathbf{y}_t^{(k)}), & \text{for } i \in \mathcal{S}_t \\ \mathbf{g}_{i,t-1}^{(k)}, & \text{otherwise} \end{cases}$$

$$\mathbf{h}_{i,t}^{(k)} = \begin{cases} \nabla_{\mathbf{y}} f_i(\mathbf{x}_t^{(k)}, \mathbf{y}_t^{(k)}), & \text{for } i \in \mathcal{S}_t \\ \mathbf{h}_{i,t-1}^{(k)}, & \text{otherwise} \end{cases}$$

7: **end for**

---

the connection among these $K$ workers. When $w_{ij} > 0$, the workers $p_i$ and $p_j$ are connected. Otherwise, they are disconnected and then cannot communicate to each other. In addition, for the adjacency matrix, we have the following assumption.

**Assumption 1.** *The adjacency matrix $W$ satisfies following properties:*

- *$W$ is nonnegative, i.e., $w_{ij} \geq 0$.*

- *$W$ is symmetric, i.e., $W^T = W$.*

- *$W$ is doubly stochastic, i.e., $W\mathbf{1} = \mathbf{1}$ and $\mathbf{1}^T W = \mathbf{1}^T$.*

- *The eigenvalues $\{\lambda_i\}_{i=1}^n$ of $W$ satisfy $|\lambda_n| \leq \cdots \leq |\lambda_2| < |\lambda_1| = 1$.*

This assumption is also used in existing works Lian et al. (2017); Koloskova et al. (2019); Liu et al. (2019b). In this paper, the spectral gap is represented by $1 - \lambda$ where $\lambda \triangleq |\lambda_2|$.

## 3.2   Method

In Algorithm 1, we developed a novel efficient descentralized stochastic gradient descent ascent (DSGDA) method. Specifically, each worker computes the stochastic gradient with its local dataset and then updates its local model parameters. In detail, at the $t$-th iteration, the $k$-th worker samples a mini-batch of samples $\mathcal{S}_t$ to compute the variance-reduced gradient regarding $\mathbf{x}$ as follows:

$$\begin{aligned} \mathbf{v}_t^{(k)} = &\frac{1}{s_t} \sum_{i \in \mathcal{S}_t} \left( \nabla_{\mathbf{x}} f_i(\mathbf{x}_t^{(k)}, \mathbf{y}_t^{(k)}) - \nabla_{\mathbf{x}} f_i(\mathbf{x}_{t-1}^{(k)}, \mathbf{y}_{t-1}^{(k)}) \right) \\ &+ (1 - \rho_t) \mathbf{v}_{t-1}^{(k)} + \rho_t \left( \frac{1}{s_t} \sum_{i \in \mathcal{S}_t} (\nabla_{\mathbf{x}} f_i(\mathbf{x}_{t-1}^{(k)}, \mathbf{y}_{t-1}^{(k)}) - \mathbf{g}_{i,t-1}^{(k)}) + \frac{1}{n} \sum_{j=1}^n \mathbf{g}_{j,t-1}^{(k)} \right), \end{aligned} \tag{2}$$

where $\rho_t \in [0, 1]$ is a hyperparameter, $\mathbf{x}_t^{(k)}$ and $\mathbf{y}_t^{(k)}$ denote the model parameters on the $k$-th worker in the $t$-th iteration, $\nabla_{\mathbf{x}} f_i(\mathbf{x}_t^{(k)}, \mathbf{y}_t^{(k)})$ denotes the stochastic gradient regarding $\mathbf{x}$, $\mathbf{v}_t^{(k)}$ is the corresponding

variance-reduced gradient, $\mathbf{g}_{i,t}^{(k)}$ stores the stochastic gradient of the $i$-th sample on the $k$-th worker, which is updated as follows:

$$\mathbf{g}_{i,t}^{(k)} = \begin{cases} \nabla_{\mathbf{x}} f_i(\mathbf{x}_t^{(k)}, \mathbf{y}_t^{(k)}), & \text{for } i \in \mathcal{S}_t \\ \mathbf{g}_{i,t-1}^{(k)}, & \text{otherwise .} \end{cases} \tag{3}$$

Similarly, to update $\mathbf{y}$, the $k$-th worker uses the same mini-batch of samples $\mathcal{S}_t$ to compute the variance-reduced gradient regarding $\mathbf{y}$ as follows:

$$\begin{aligned} \mathbf{u}_t^{(k)} = \frac{1}{s_t} \sum_{i \in \mathcal{S}_t} \left( \nabla_{\mathbf{y}} f_i(\mathbf{x}_t^{(k)}, \mathbf{y}_t^{(k)}) - \nabla_{\mathbf{y}} f_i(\mathbf{x}_{t-1}^{(k)}, \mathbf{y}_{t-1}^{(k)}) \right) \\ + (1 - \rho_t)\mathbf{u}_{t-1}^{(k)} + \rho_t \left( \frac{1}{s_t} \sum_{i \in \mathcal{S}_t} (\nabla_{\mathbf{y}} f_i(\mathbf{x}_{t-1}^{(k)}, \mathbf{y}_{t-1}^{(k)}) - \mathbf{h}_{i,t-1}^{(k)}) + \frac{1}{n} \sum_{j=1}^{n} \mathbf{h}_{j,t-1}^{(k)} \right) , \end{aligned} \tag{4}$$

where $\mathbf{u}_t^{(k)}$ is the variance-reduced gradient for the variable $\mathbf{y}$, $\mathbf{h}_{i,t}^{(k)}$ stores the stochastic gradient of the $i$-th sample on the $k$-th worker for the variable $\mathbf{y}$. Similar to $\mathbf{g}_{i,t}^{(k)}$, $\mathbf{h}_{i,t}^{(k)}$ is updated as follows:

$$\mathbf{h}_{i,t}^{(k)} = \begin{cases} \nabla_{\mathbf{y}} f_i(\mathbf{x}_t^{(k)}, \mathbf{y}_t^{(k)}), & \text{for } i \in \mathcal{S}_t \\ \mathbf{h}_{i,t-1}^{(k)}, & \text{otherwise .} \end{cases} \tag{5}$$

After obtaining the variance-reduced gradient, the $k$-th worker employs the gradient tracking communication scheme to communicate with its neighboring workers:

$$\mathbf{a}_t^{(k)} = \sum_{j \in \mathcal{N}_k} w_{kj} \mathbf{a}_{t-1}^{(j)} + \mathbf{v}_t^{(k)} - \mathbf{v}_{t-1}^{(k)} , \mathbf{b}_t^{(k)} = \sum_{j \in \mathcal{N}_k} w_{kj} \mathbf{b}_{t-1}^{(j)} + \mathbf{u}_t^{(k)} - \mathbf{u}_{t-1}^{(k)} , \tag{6}$$

where $\mathcal{N}_k$ is the neighboring workers of the $k$-th worker, $\mathbf{a}_t^{(k)}$ and $\mathbf{b}_t^{(k)}$ are the gradients after communicating with the neighboring workers.

Then, the $k$-th worker updates its local model parameter $\mathbf{x}_t^{(k)}$ as follows:

$$\mathbf{x}_{t+\frac{1}{2}}^{(k)} = \sum_{j \in \mathcal{N}_k} w_{kj} \mathbf{x}_t^{(j)} - \gamma_1 \mathbf{a}_t^{(k)} , \quad \mathbf{x}_{t+1}^{(k)} = \mathbf{x}_t^{(k)} + \eta(\mathbf{x}_{t+\frac{1}{2}}^{(k)} - \mathbf{x}_t^{(k)}) , \tag{7}$$

where $\gamma_1 > 0$ and $0 < \eta < 1$ are two hyperparameters. Similarly, $\mathbf{y}_t^{(k)}$ is also updated as follows:

$$\mathbf{y}_{t+\frac{1}{2}}^{(k)} = \sum_{j \in \mathcal{N}_k} w_{kj} \mathbf{y}_t^{(j)} + \gamma_2 \mathbf{b}_t^{(k)} , \quad \mathbf{y}_{t+1}^{(k)} = \mathbf{y}_t^{(k)} + \eta(\mathbf{y}_{t+\frac{1}{2}}^{(k)} - \mathbf{y}_t^{(k)}) , \tag{8}$$

where $\gamma_2 > 0$ and $0 < \eta < 1$ are two hyperparameters.

All workers in the decentralized training system repeat the aforementioned steps to update $\mathbf{x}$ and $\mathbf{y}$ until it converges.

The variance-reduced gradient estimator in Eq. (2) was first proposed in Li & Richtárik (2021). But they only focus on the minimization problem and ignore the decentralized setting. In fact, it is nontrivial to apply this variance-reduced gradient estimator to the decentralized minimax problem. Especially, it is challenging to establish the convergence rate, which is shown in the next section.

## 4 Theoretical Analysis

### 4.1 Convergence Rate

To establish the convergence rate of Algorithm 1, we introduce the following assumptions, which are commonly used in existing works.

**Assumption 2.** *(Smoothness) Each function $f_i^{(k)}(\cdot, \cdot)$ is L-smooth. i.e., for any $(\mathbf{x_1}, \mathbf{y_1})$ and $(\mathbf{x_2}, \mathbf{y_2})$, there exists $L > 0$, such that*

$$\|\nabla f_i^{(k)}(\mathbf{x_1}, \mathbf{y_1}) - \nabla f_i^{(k)}(\mathbf{x_2}, \mathbf{y_2})\|^2 \leq L^2 \|\mathbf{x_1} - \mathbf{x_2}\|^2 + L^2 \|\mathbf{y_1} - \mathbf{y_2}\|^2 . \tag{9}$$

**Assumption 3.** *(Strong concavity) The function $f^{(k)}(\mathbf{x}, \mathbf{y})$ is $\mu$-strongly concave with respect to $\mathbf{y}$, i.e., for any $(\mathbf{x}, \mathbf{y_1})$ and $(\mathbf{x}, \mathbf{y_2})$, there exists $\mu > 0$, such that*

$$f^{(k)}(\mathbf{x}, \mathbf{y_1}) \leq f^{(k)}(\mathbf{x}, \mathbf{y_2}) + \langle \nabla_{\mathbf{y}} f^{(k)}(\mathbf{x}, \mathbf{y_2}), \mathbf{y_1} - \mathbf{y_2} \rangle - \frac{\mu}{2} \|\mathbf{y_1} - \mathbf{y_2}\|^2 . \tag{10}$$

Here, we denote the condition number by $\kappa = L/\mu$. Throughout this paper, we denote $\bar{\mathbf{c}} = \frac{1}{K} \sum_{k=1}^K \mathbf{c}_k$, where $\mathbf{c}_k$ is the variable on the $k$-th worker. Based on these assumptions, we establish the convergence rate of our method for nonconvex-strongly-concave problems in Theorem 1.

**Theorem 1.** *Given Assumptions 1-3, if setting $\gamma_1 \leq \min\{\gamma_{1,1}, \gamma_{1,2}, \gamma_{1,3}\}$, $\gamma_2 \leq \min\{\gamma_{2,1}, \gamma_{2,2}, \gamma_{2,3}\}$ where*

$$\gamma_{1,1} \leq \frac{(1-\lambda)^2}{\sqrt{12}\rho_1 \kappa L} \Big/ \sqrt{212 + 3\Big(\frac{1019}{s_1\rho_1} + \frac{4104\rho_1 n^2}{s_1^3}\Big)} , \quad \gamma_{1,2} \leq \frac{1}{8\kappa L} \Big/ \sqrt{\frac{1019}{s_1\rho_1} + \frac{4104\rho_1 n^2}{s_1^3}} , \quad \gamma_{1,3} \leq \frac{\gamma_2}{20\kappa^2} ,$$

$$\gamma_{2,1} \leq \frac{3}{2\kappa L} \Big/ \Big(\frac{1019}{s_1\rho_1} + \frac{(4104\rho_1 n^2)}{s_1^3}\Big) , \gamma_{2,2} \leq \frac{1}{6L} , \quad \gamma_{2,3} \leq \frac{(1-\lambda)^2}{\sqrt{2}\rho_1 L} \Big/ \sqrt{212 + 3\Big(\frac{1019}{s_1\rho_1} + \frac{4104\rho_1 n^2}{s_1^3}\Big)} , \tag{11}$$

*and $\eta < \min\{1, \frac{1}{2\gamma_1 L_\Phi}, \frac{1}{\sqrt{12}} \Big/ \sqrt{\frac{1019}{s_1\rho_1} + \frac{4104\rho_1 n^2}{s_1^3}}\}$, $s_t = s_1$ for $t > 0$, $\rho_t = \rho_1 = \frac{s_1}{2n}$ for $t > 0$, $\rho_0 = 1$, our algorithm is able to achieve the following convergence rate:*

$$\frac{1}{T} \sum_{t=0}^{T-1} (\mathbb{E}[\|\nabla \Phi(\bar{\mathbf{x}}_t)\|^2] + L^2 \mathbb{E}[\|\bar{\mathbf{y}}_t - \mathbf{y}^*(\bar{\mathbf{x}}_t)\|^2]) \leq \frac{2(\Phi(\bar{\mathbf{x}}_0) - \Phi(\mathbf{x}_*))}{\gamma_1 \eta T} + \frac{12\kappa L}{\gamma_2 \eta T} \mathbb{E}[\|\bar{\mathbf{y}}_0 - \mathbf{y}^*(\bar{\mathbf{x}}_0)\|^2]$$

$$+ \left(\frac{4\kappa^2 (n - s_0)}{s_0 s_1}\Big(7 + 15369 L^2 \gamma_1^2 + \frac{424\gamma_1^2 L^2}{(1-\lambda^2)^2}\Big) + \frac{28(s_0 - s_0^2/n)}{2s_0 s_1}\right) \frac{1}{TK} \sum_{k=1}^K \frac{1}{n} \sum_{i=1}^n \mathbb{E}[\|\nabla_{\mathbf{x}} f_i(\mathbf{x}_0^{(k)}, \mathbf{y}_0^{(k)})\|^2]$$

$$+ \left(\frac{4\kappa^2 (n - s_0)}{s_0 s_1}\Big(258 + 50L^2 + 15369 L^2 \gamma_2^2 + \frac{424\gamma_2^2 L^2}{(1-\lambda^2)^2}\Big) + \frac{2024\kappa^2(s_0 - s_0^2/n)}{2s_0 s_1}\right) \frac{1}{TK} \sum_{k=1}^K \frac{1}{n} \sum_{i=1}^n \mathbb{E}[\|\nabla_{\mathbf{y}} f_i(\mathbf{x}_0^{(k)}, \mathbf{y}_0^{(k)})\|^2] , \tag{12}$$

*where $\mathbf{y}^*(\bar{\mathbf{x}}) = \arg\max_{\mathbf{y}} \frac{1}{K} \sum_{k=1}^K f^{(k)}(\bar{\mathbf{x}}, \mathbf{y})$, $\Phi(\mathbf{x}) = \frac{1}{K} \sum_{k=1}^K \Phi^{(k)}(\mathbf{x}) = \frac{1}{K} \sum_{k=1}^K f^{(k)}(\mathbf{x}, \mathbf{y}^*(\mathbf{x}))$ and it is $L_\Phi = 2\kappa L$ smooth.*

**Corollary 1.** *Given Assumptions 1-3, by setting $s_0 = s_1 = \sqrt{n}$, $\rho_1 = \frac{s_1}{2n}$, we can get $\gamma_1 = O((1-\lambda)^2/\kappa^3)$, $\gamma_2 = O((1-\lambda)^2/\kappa)$, and $\eta = O(1)$ under the worst case. Then, by setting $T = O(\frac{\kappa^3}{(1-\lambda)^2\epsilon^2})$, our algorithm can achieve the $\epsilon$-accuracy solution:*

$$\frac{1}{T} \sum_{t=0}^{T-1} (\mathbb{E}[\|\nabla \Phi(\bar{\mathbf{x}}_t)\|^2] + L^2 \mathbb{E}[\|\bar{\mathbf{y}}_t - \mathbf{y}^*(\bar{\mathbf{x}}_t)\|^2]) \leq \epsilon^2 . \tag{13}$$

**Remark 1.** *From Corollary 1, it is easy to know that the communication complexity of our method is $O(\frac{\kappa^3}{(1-\lambda)^2\epsilon^2})$ and the sample complexity is $T \times \sqrt{n} = O(\frac{\sqrt{n}\kappa^3}{(1-\lambda)^2\epsilon^2})$.*

**Remark 2.** *Compared with Xian et al. (2021) whose step sizes are $O((1-\lambda)^2\epsilon/\kappa^3)$ and $O((1-\lambda)^2\epsilon/\kappa)$, our step sizes, i.e., $\gamma_1$, $\gamma_2$, and $\eta$, are independent of $\epsilon$. In addition, our communication complexity is better than $O(\frac{\kappa^3}{(1-\lambda)^2\epsilon^3})$ of Xian et al. (2021).*

**Remark 3.** *The step sizes of Zhang et al. (2021b) are also independent of $\epsilon$ and have the same order dependence on the spectral gap and condition number as our $\gamma_1$ and $\gamma_2$. But its sample complexity $O(n + \frac{\sqrt{n}\kappa^3}{(1-\lambda)^2\epsilon^2})$ is worse than ours because it needs to periodically compute the full gradient.*

### 4.2 Proof Sketch

In this subsection, we present the proof sketch of our Theorem 1. The detailed proof can be found in supplementary materials.

To investigate the convergence rate of our method, we propose a novel potential function as follows:

$$
\begin{aligned}
H_t =\ & \mathbb{E}[\Phi(\bar{\mathbf{x}}_t)] + \frac{6\gamma_1 L^2}{\gamma_2 \mu} \mathbb{E}[\|\bar{\mathbf{y}}_t - \mathbf{y}^*(\bar{\mathbf{x}}_t)\|^2] + \frac{106\gamma_1 \kappa^2 L^2}{(1-\lambda^2)} \frac{1}{K} \sum_{k=1}^{K} \mathbb{E}[\|\bar{\mathbf{x}}_t - \mathbf{x}_t^{(k)}\|^2] \\
& + \frac{(1-\lambda^2)\gamma_1 \eta}{6\rho_1} \frac{1}{K} \sum_{k=1}^{K} \mathbb{E}[\|\bar{\mathbf{a}}_t - \mathbf{a}_t^{(k)}\|^2] + \frac{(1-\lambda^2)\eta\gamma_1 L^2}{\rho_1 \mu^2} \frac{1}{K} \sum_{k=1}^{K} \mathbb{E}[\|\bar{\mathbf{b}}_t - \mathbf{b}_t^{(k)}\|^2] \\
& + \frac{3\gamma_1 \eta}{\rho_1} \frac{1}{K} \sum_{k=1}^{K} \mathbb{E}[\|\nabla_{\mathbf{x}} f(\mathbf{x}_t^{(k)}, \mathbf{y}_t^{(k)}) - \mathbf{v}_t^{(k)}\|^2] + \frac{250\eta\gamma_1 L^2}{\rho_1 \mu^2} \frac{1}{K} \sum_{k=1}^{K} \mathbb{E}[\|\nabla_{\mathbf{y}} f(\mathbf{x}_t^{(k)}, \mathbf{y}_t^{(k)}) - \mathbf{u}_t^{(k)}\|^2] \\
& + \frac{14 n \rho_1 \gamma_1 \eta}{s_1^2} \frac{1}{K} \sum_{k=1}^{K} \mathbb{E}[\frac{1}{n} \sum_{j=1}^{n} \|\nabla_{\mathbf{x}} f_j(\mathbf{x}_t^{(k)}, \mathbf{y}_t^{(k)}) - \mathbf{g}_{j,t}^{(k)}\|^2] + \frac{106\gamma_1 \kappa^2 L^2}{(1-\lambda^2)} \frac{1}{K} \sum_{k=1}^{K} \mathbb{E}[\|\bar{\mathbf{y}}_t - \mathbf{y}_t^{(k)}\|^2] \\
& + \frac{1012 n \rho_1 \eta \gamma_1 L^2}{s_1^2 \mu^2} \frac{1}{K} \sum_{k=1}^{K} \mathbb{E}[\frac{1}{n} \sum_{j=1}^{n} \|\nabla_{\mathbf{y}} f_j(\mathbf{x}_t^{(k)}, \mathbf{y}_t^{(k)}) - \mathbf{h}_{j,t}^{(k)}\|^2] \ .
\end{aligned}
\tag{14}
$$

In the potential function $H_t$, $\frac{1}{K} \sum_{k=1}^{K} \|\bar{\mathbf{x}}_t - \mathbf{x}_t^{(k)}\|^2$, $\frac{1}{K} \sum_{k=1}^{K} \|\bar{\mathbf{y}}_t - \mathbf{y}_t^{(k)}\|^2$, $\frac{1}{K} \sum_{k=1}^{K} \|\bar{\mathbf{a}}_t - \mathbf{a}_t^{(k)}\|^2$, and $\frac{1}{K} \sum_{k=1}^{K} \|\bar{\mathbf{b}}_t - \mathbf{b}_t^{(k)}\|^2$ are the consensus error with respect to the variables and tracked gradients. The last four terms characterise the gradient variance. Then, we can investigate how the potential function evolves across iterations by studying each item in this potential function. In particular, we can get

$$
H_{t+1} - H_t \leq -\frac{\gamma_1 \eta}{2} \mathbb{E}[\|\nabla \Phi(\bar{\mathbf{x}}_t)\|^2] - \frac{\gamma_1 \eta L^2}{2} \mathbb{E}[\|\bar{\mathbf{y}}_t - \mathbf{y}^*(\bar{\mathbf{x}}_t)\|^2] \ .
\tag{15}
$$

Then, based on this inequality, we can complete the proof. The detailed proof can be found in Supplementary Materials.

## 5 Experiments

In this section, we present experimental results to demonstrate the empirical performance of our method.

### 5.1 AUC Maximization

AUC maximization is a commonly used method for the imbalanced data classification problem. Recently, Ying et al. (2016) reformulated the AUC maximization problem as an minimax optimization problem to facilitate stochastic training for large-scale data. In our experiment, we employ our method to optimize the AUC maximization problem, which is defined as follows:

$$
\begin{aligned}
\min_{\boldsymbol{\theta}, \hat{\theta}_1, \hat{\theta}_2} \max_{\tilde{\theta}} \ell(\boldsymbol{\theta}, \hat{\theta}_1, \hat{\theta}_2, \tilde{\theta}) =\ & (1-p)(\boldsymbol{\theta}^T \mathbf{a} - \hat{\theta}_1)^2 \mathbb{I}_{[b=1]} + p(\boldsymbol{\theta}^T \mathbf{a} - \hat{\theta}_2)^2 \mathbb{I}_{[b=-1]} \\
& + 2(1+\tilde{\theta})(p\boldsymbol{\theta}^T \mathbf{a} \mathbb{I}_{[b=-1]} - (1-p)\boldsymbol{\theta}^T \mathbf{a} \mathbb{I}_{[b=1]}) - p(1-p)\tilde{\theta}^2 + \gamma \sum_{j=1}^{d} \frac{\boldsymbol{\theta}_j^2}{1+\boldsymbol{\theta}_j^2} \ ,
\end{aligned}
\tag{16}
$$

where $\boldsymbol{\theta} \in \mathbb{R}^d$ denotes the model parameter of the classifier, $\hat{\theta}_1 \in \mathbb{R}$, $\hat{\theta}_2 \in \mathbb{R}$ are two auxiliary model parameters for the minimization subproblem, $\tilde{\theta} \in \mathbb{R}$ is the model parameter for the maximization subproblem, $\{\mathbf{a}, b\}$ denotes training samples, $\gamma > 0$ is a hyperparameter for the regularization term. In our experiments, we set $\gamma$ to 0.001. Obviously, this is a nonconvex-strongly-concave optimization problem. Then, we can use our Algorithm 1 to optimize this problem.

## 5.2 Experimental Settings

In our experiments, we employ three binary classification datasets: a9a, w8a, and ijcnn1. All of them are imbalanced datasets. The detailed information about these datasets can be found in LIBSVM [3]. We randomly select 20% samples as the testing set and the left samples as the training set. Throughout our experiments, we employ ten workers. Then, the training samples are randomly distributed to all workers.

To evaluate the performance of our method, we compare it with the state-of-the-art decentralized optimization algorithm: DM-HSGD Xian et al. (2021), GT-SRVR Zhang et al. (2021b), and GT-SRVRI Zhang et al. (2021b). As for their step sizes, we set them in terms of Remarks 2 and 3. Specifically, since we employed the Erdos-Renyi random graph with the edge probability being 0.5 to generate the communication network, whose spectral gap is in the order of $O(1)$ Ying et al. (2021), we assume the spectral gap as 0.5. Additionally, the solution accuracy $\epsilon$ is set to 0.01. Then, the step sizes of two variables of DM-HSGD are set to $(1 - \lambda)^2 \epsilon = 0.5^2 \times 0.01$ and the coefficient for the variance-reduced gradient estimator is set to $\epsilon \min\{1, n\epsilon\} = 0.01 \times 10 \times 0.01$ according to Theorem 1 in Xian et al. (2021). As for GT-SRVR and GT-SRVRI, the step sizes of two variables are set to $0.5^2$ based on Remark 3 [4]. As for our method, the step sizes $\gamma_1$ and $\gamma_2$ are set to $(1 - \lambda)^2 = 0.5^2$ in terms of Remark 1. Note that we omit the condition number for all step sizes since they are the same for all methods and it is difficult to obtain. As for our method, since $\eta$ is independent of the spectral gap and the solution accuracy, we set it to 0.9 throughout our experiments. Additionally, $\rho_1$ is set to $0.5/\sqrt{n}$. Moreover, the batch size is set to $\sqrt{n}$ for GT-SRVR and our method according to Corollary 1, and that is set to 64 for DM-HSGD.

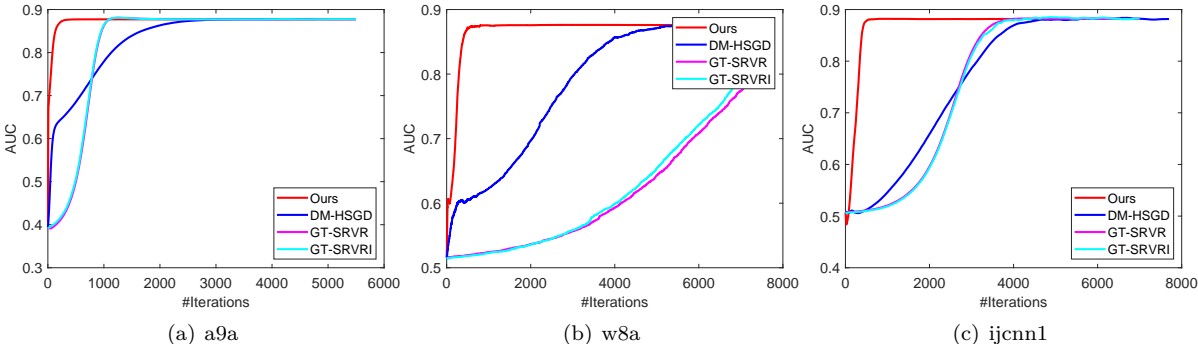

Figure 1: The test AUC versus the number of iterations when using the random communication graph.

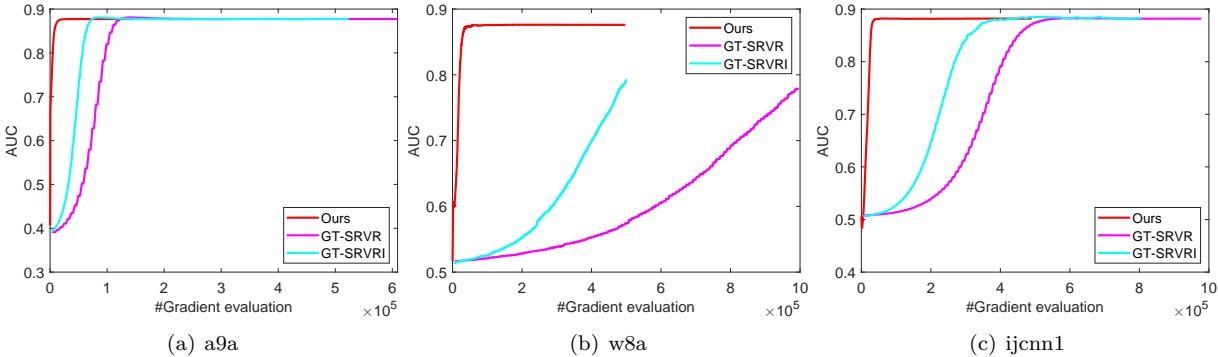

Figure 2: The test AUC versus the number of gradient evaluation when using the random communication graph.

---

[3] https://www.csie.ntu.edu.tw/~cjlin/libsvmtools/datasets/
[4] Since GT-SRVR's theoretical step size leads to divergence for a9a dataset, we scaled it by 0.01 for the random graph and 0.5 for the line graph, respectively.

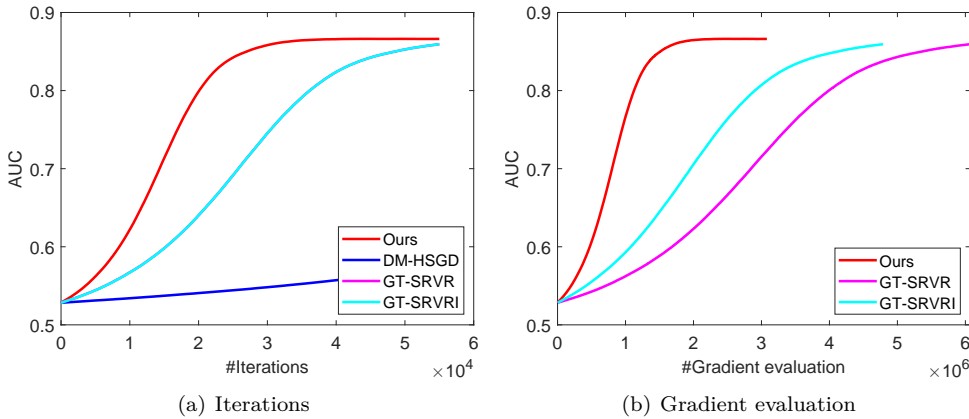

(a) Iterations

(b) Gradient evaluation

Figure 3: The test AUC when using the line communication graph for a9a dataset.

## 5.3 Experimental Results

To compare our method with baseline methods, we plot test AUC versus the number of iterations in Figure 1 and that versus the number of gradient evaluation in Figure 2 [5]. From Figure 1, it can be observed that our method converges much faster than all baseline methods in terms of the number of iterations, which confirms the efficacy of our method. Additionally, from Figure 2, it can be observed that our method converges faster than GT-SRVR and GT-SRVRI in terms of the number of gradient evaluation, which confirms the sample complexity of our method is much smaller than theirs.

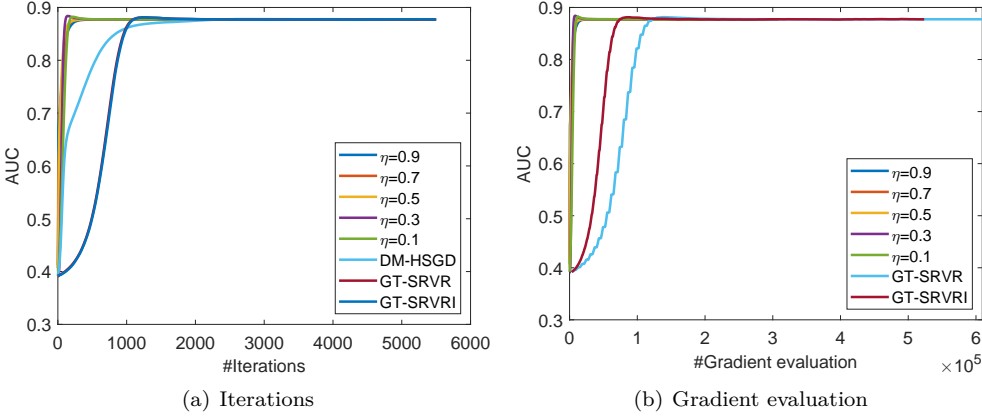

(a) Iterations

(b) Gradient evaluation

Figure 4: The test AUC on different $\eta$ when using the random communication graph for a9a dataset.

Moreover, we verify the performance of our method on the line communication network. In particular, in a line communication network, each worker connects with only two neighboring workers. Its spectral gap is in the order of $O(1/K^2)$ Ying et al. (2021) where $K = 10$ in our experiments. Accordingly, we set the step size to $(1 - \lambda)^2 \epsilon = 0.01^2 \times 0.01$ for DM-HSGD, $(1 - \lambda)^2 = 0.01^2$ for other methods. The other hyperparameters are the same with the random communication network. In Figure 3, we plot the test AUC score versus the number of iterations and the number of gradient evaluation for a9a dataset. It can be observed that our method still converges much faster than two baseline methods, which further confirms the efficacy of our method.

---

[5]We didn't plot the gradient evaluation of DM-HSGD since it is for the stochastic setting, rather than the finite-sum setting.

Finally, we demonstrate the performance of our method with different values of $\eta$ in Figure 4. It can be observed that our method with different $\eta$ still converges much faster than baseline methods in terms of the number of iterations and gradient evaluation.

## 6    Conclusion

In this paper, we developed a novel decentralized stochastic gradient descent ascent method for the finite-sum optimization problems. We also established the convergence rate of our method, providing the sample complexity and communication complexity. Importantly, our method can achieve the better communication or computation complexity than existing decentralized methods. Finally, the extensive experimental results confirm the efficacy of our method.

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

## A  Appendix

Throughout this paper, we denote $C_t = [\mathbf{c}_t^{(1)}, \cdots, \mathbf{c}_t^{(K)}]$ and $\bar{C}_t = [\bar{\mathbf{c}}_t, \cdots, \bar{\mathbf{c}}_t]$ where $[\mathbf{c}_t^{(k)}$ denotes the variable on the $k$-th device in the $t$-th iteration.

**Lemma 1.** *Given Assumption 2-3, by setting* $\eta \leq \frac{1}{2\gamma_1 L_\Phi}$, *we have*

$$\Phi(\bar{\mathbf{x}}_{t+1}) \leq \Phi(\bar{\mathbf{x}}_t) - \frac{\gamma_1\eta}{2}\|\nabla\Phi(\bar{\mathbf{x}}_t)\|^2 - \frac{\gamma_1\eta}{4}\|\bar{\mathbf{v}}_t\|^2 + \gamma_1\eta L^2\|\bar{\mathbf{y}}_t - \mathbf{y}^*(\bar{\mathbf{x}}_t)\|^2 + \frac{2\gamma_1\eta L^2}{K}\sum_{k=1}^{K}\|\bar{\mathbf{x}}_t - \mathbf{x}_t^{(k)}\|^2$$

$$+ \frac{2\gamma_1\eta L^2}{K}\sum_{k=1}^{K}\|\bar{\mathbf{y}}_t - \mathbf{y}_t^{(k)}\|^2 + \frac{2\gamma_1\eta}{K}\sum_{k=1}^{K}\|\nabla_{\mathbf{x}}f(\mathbf{x}_t^{(k)}, \mathbf{y}_t^{(k)}) - \mathbf{v}_t^{(k)}\|^2 . \tag{17}$$

*Proof.*

$$\Phi(\bar{\mathbf{x}}_{t+1}) \leq \Phi(\bar{\mathbf{x}}_t) + \langle\nabla\Phi(\bar{\mathbf{x}}_t), \bar{\mathbf{x}}_{t+1} - \bar{\mathbf{x}}_t\rangle + \frac{L_\Phi}{2}\|\bar{\mathbf{x}}_{t+1} - \bar{\mathbf{x}}_t\|^2$$

$$= \Phi(\bar{\mathbf{x}}_t) - \gamma_1\eta\langle\nabla\Phi(\bar{\mathbf{x}}_t), \bar{\mathbf{v}}_t\rangle + \frac{\gamma_1^2\eta^2 L_\Phi}{2}\|\bar{\mathbf{v}}_t\|^2$$

$$= \Phi(\bar{\mathbf{x}}_t) - \frac{\gamma_1\eta}{2}\|\nabla\Phi(\bar{\mathbf{x}}_t)\|^2 + \left(\frac{\gamma_1^2\eta^2 L_\Phi}{2} - \frac{\gamma_1\eta}{2}\right)\|\bar{\mathbf{v}}_t\|^2 + \frac{\gamma_1\eta}{2}\|\nabla\Phi(\bar{\mathbf{x}}_t) - \bar{\mathbf{v}}_t\|^2$$

$$\leq \Phi(\bar{\mathbf{x}}_t) - \frac{\gamma_1\eta}{2}\|\nabla\Phi(\bar{\mathbf{x}}_t)\|^2 + \left(\frac{\gamma_1^2\eta^2 L_\Phi}{2} - \frac{\gamma_1\eta}{2}\right)\|\bar{\mathbf{v}}_t\|^2$$

$$+ \gamma_1\eta\|\nabla\Phi(\bar{\mathbf{x}}_t) - \frac{1}{K}\sum_{k=1}^{K}\nabla_{\mathbf{x}}f^{(k)}(\bar{\mathbf{x}}_t, \bar{\mathbf{y}})\|^2 + \gamma_1\eta\|\frac{1}{K}\sum_{k=1}^{K}\nabla_{\mathbf{x}}f^{(k)}(\bar{\mathbf{x}}_t, \bar{\mathbf{y}}) - \bar{\mathbf{v}}_t\|^2$$

$$\leq \Phi(\bar{\mathbf{x}}_t) - \frac{\gamma_1\eta}{2}\|\nabla\Phi(\bar{\mathbf{x}}_t)\|^2 + \left(\frac{\gamma_1^2\eta^2 L_\Phi}{2} - \frac{\gamma_1\eta}{2}\right)\|\bar{\mathbf{v}}_t\|^2 \tag{18}$$

$$+ \gamma_1\eta\|\nabla\Phi(\bar{\mathbf{x}}_t) - \frac{1}{K}\sum_{k=1}^{K}\nabla_{\mathbf{x}}f^{(k)}(\bar{\mathbf{x}}_t, \bar{\mathbf{y}})\|^2 + 2\gamma_1\eta\|\frac{1}{K}\sum_{k=1}^{K}\nabla_{\mathbf{x}}f^{(k)}(\bar{\mathbf{x}}_t, \bar{\mathbf{y}}) - \frac{1}{K}\sum_{k=1}^{K}\nabla_{\mathbf{x}}f(\mathbf{x}_t^{(k)}, \mathbf{y}_t^{(k)})\|^2$$

$$+ 2\gamma_1\eta\|\frac{1}{K}\sum_{k=1}^{K}\nabla_{\mathbf{x}}f(\mathbf{x}_t^{(k)}, \mathbf{y}_t^{(k)}) - \bar{\mathbf{v}}_t\|^2$$

$$\leq \Phi(\bar{\mathbf{x}}_t) - \frac{\gamma_1\eta}{2}\|\nabla\Phi(\bar{\mathbf{x}}_t)\|^2 - \frac{\gamma_1\eta}{4}\|\bar{\mathbf{v}}_t\|^2 + \gamma_1\eta L^2\|\bar{\mathbf{y}}_t - \mathbf{y}^*(\bar{\mathbf{x}}_t)\|^2$$

$$+ \frac{2\gamma_1\eta L^2}{K}\sum_{k=1}^{K}\|\bar{\mathbf{x}}_t - \mathbf{x}_t^{(k)}\|^2 + \frac{2\gamma_1\eta L^2}{K}\sum_{k=1}^{K}\|\bar{\mathbf{y}}_t - \mathbf{y}_t^{(k)}\|^2 + \frac{2\gamma_1\eta}{K}\sum_{k=1}^{K}\|\nabla_{\mathbf{x}}f(\mathbf{x}_t^{(k)}, \mathbf{y}_t^{(k)}) - \mathbf{v}_t^{(k)}\|^2 ,$$

where the last inequality holds due to $\eta \leq \frac{1}{2\gamma_1 L_\Phi}$ and the following inequality.

$$\|\nabla\Phi(\bar{\mathbf{x}}_t) - \frac{1}{K}\sum_{k=1}^{K}\nabla_{\mathbf{x}}f^{(k)}(\bar{\mathbf{x}}_t, \bar{\mathbf{y}})\|^2$$

$$= \|\frac{1}{K}\sum_{k=1}^{K}\nabla_{\mathbf{x}}f^{(k)}(\bar{\mathbf{x}}_t, \mathbf{y}^*(\bar{\mathbf{x}}_t)) - \frac{1}{K}\sum_{k=1}^{K}\nabla_{\mathbf{x}}f^{(k)}(\bar{\mathbf{x}}_t, \bar{\mathbf{y}})\|^2 \tag{19}$$

$$\leq \frac{L^2}{K}\sum_{k=1}^{K}(\|\bar{\mathbf{x}}_t - \bar{\mathbf{x}}_t\|^2 + \|\mathbf{y}^*(\bar{\mathbf{x}}_t) - \bar{\mathbf{y}}\|^2)$$

$$= L^2\|\mathbf{y}^*(\bar{\mathbf{x}}_t) - \bar{\mathbf{y}}\|^2 .$$

$\square$

**Lemma 2.** *Given Assumption 2-3, by setting $\gamma_2 \leq \frac{1}{6L}$ and $\eta < 1$, we have*

$$
\|\bar{\mathbf{y}}_{t+1} - \mathbf{y}^*(\bar{\mathbf{x}}_{t+1})\|^2 \leq (1 - \frac{\eta\gamma_2\mu}{4})\|\bar{\mathbf{y}}_t - \mathbf{y}^*(\bar{\mathbf{x}}_t)\|^2
$$

$$
- \frac{3\gamma_2^2\eta^2}{4}\|\bar{\mathbf{u}}_t\|^2 + \frac{25\eta\gamma_2 L^2}{3\mu}\frac{1}{K}\sum_{k=1}^{K}\|\bar{\mathbf{y}}_t - \mathbf{y}_t^{(k)}\|^2
$$

$$
+ \frac{25\eta\gamma_1^2\kappa^2}{6\gamma_2\mu}\|\bar{\mathbf{v}}_t\|^2 + \frac{25\eta\gamma_2 L^2}{3\mu}\frac{1}{K}\sum_{k=1}^{K}\|\bar{\mathbf{x}}_t - \mathbf{x}_t^{(k)}\|^2 \tag{20}
$$

$$
+ \frac{25\eta\gamma_2}{3\mu}\frac{1}{K}\sum_{k=1}^{K}\|\nabla_{\mathbf{y}}f^{(k)}(\mathbf{x}_t^{(k)}, \mathbf{y}_t^{(k)}) - \mathbf{u}_t^{(k)}\|^2 \ .
$$

*Proof.* By setting $\gamma_2 \leq \frac{1}{6L}$ and $\eta < 1$, we have

$$
\|\bar{\mathbf{y}}_{t+1} - \mathbf{y}^*(\bar{\mathbf{x}}_{t+1})\|^2
$$

$$
\leq (1 - \frac{\eta\gamma_2\mu}{4})\|\bar{\mathbf{y}}_t - \mathbf{y}^*(\bar{\mathbf{x}}_t)\|^2 - \frac{3\gamma_2^2\eta}{4}\|\bar{\mathbf{u}}_t\|^2 + \frac{25\eta\gamma_1^2\kappa^2}{6\gamma_2\mu}\|\bar{\mathbf{v}}_t\|^2 + \frac{25\eta\gamma_2}{6\mu}\|\nabla_{\mathbf{y}}f(\bar{\mathbf{x}}_t, \bar{\mathbf{y}}_t) - \bar{\mathbf{u}}_t\|^2
$$

$$
\leq (1 - \frac{\eta\gamma_2\mu}{4})\|\bar{\mathbf{y}}_t - \mathbf{y}^*(\bar{\mathbf{x}}_t)\|^2 - \frac{3\gamma_2^2\eta}{4}\|\bar{\mathbf{u}}_t\|^2 + \frac{25\eta\gamma_1^2\kappa^2}{6\gamma_2\mu}\|\bar{\mathbf{v}}_t\|^2
$$

$$
+ \frac{25\eta\gamma_2}{3\mu}\|\nabla_{\mathbf{y}}f(\bar{\mathbf{x}}_t, \bar{\mathbf{y}}_t) - \frac{1}{K}\sum_{k=1}^{K}\nabla_{\mathbf{y}}f^{(k)}(\mathbf{x}_t^{(k)}, \mathbf{y}_t^{(k)})\|^2
$$

$$
+ \frac{25\eta\gamma_2}{3\mu}\frac{1}{K}\sum_{k=1}^{K}\|\nabla_{\mathbf{y}}f^{(k)}(\mathbf{x}_t^{(k)}, \mathbf{y}_t^{(k)}) - \mathbf{u}_t^{(k)}\|^2 \tag{21}
$$

$$
\leq (1 - \frac{\eta\gamma_2\mu}{4})\|\bar{\mathbf{y}}_t - \mathbf{y}^*(\bar{\mathbf{x}}_t)\|^2 - \frac{3\gamma_2^2\eta}{4}\|\bar{\mathbf{u}}_t\|^2 + \frac{25\eta\gamma_1^2\kappa^2}{6\gamma_2\mu}\|\bar{\mathbf{v}}_t\|^2
$$

$$
+ \frac{25\eta\gamma_2 L^2}{3\mu}\frac{1}{K}\sum_{k=1}^{K}\|\bar{\mathbf{x}}_t - \mathbf{x}_t^{(k)}\|^2 + \frac{25\eta\gamma_2 L^2}{3\mu}\frac{1}{K}\sum_{k=1}^{K}\|\bar{\mathbf{y}}_t - \mathbf{y}_t^{(k)}\|^2
$$

$$
+ \frac{25\eta\gamma_2}{3\mu}\frac{1}{K}\sum_{k=1}^{K}\|\nabla_{\mathbf{y}}f^{(k)}(\mathbf{x}_t^{(k)}, \mathbf{y}_t^{(k)}) - \mathbf{u}_t^{(k)}\|^2 \ ,
$$

where the first step follows from (Huang et al., 2020a).

$\square$

**Lemma 3.** *Given Assumption 1-3, for $t > 0$, by setting $s_t = s_1$, $\rho_t = \rho_1$, we have*

$$
\sum_{k=1}^{K}\mathbb{E}[\|\mathbf{a}_{t+1}^{(k)} - \bar{\mathbf{a}}_{t+1}\|^2] \leq \frac{1+\lambda^2}{2}\sum_{k=1}^{K}\mathbb{E}[\|\mathbf{a}_t^{(k)} - \bar{\mathbf{a}}_t\|^2]
$$

$$
+ \frac{6L^2}{(1-\lambda^2)s_1}\sum_{k=1}^{K}\mathbb{E}[\|\mathbf{x}_{t+1}^{(k)} - \mathbf{x}_t^{(k)}\|^2]
$$

$$
+ \frac{6L^2}{(1-\lambda^2)s_1}\sum_{k=1}^{K}\mathbb{E}[\|\mathbf{y}_{t+1}^{(k)} - \mathbf{y}_t^{(k)}\|^2] \tag{22}
$$

$$
+ \frac{6\rho_1^2}{1-\lambda^2}\sum_{k=1}^{K}\mathbb{E}[\|\mathbf{v}_t^{(k)} - \nabla_{\mathbf{x}}f(\mathbf{x}_t^{(k)}, \mathbf{y}_t^{(k)})\|^2]
$$

$$
+ \frac{6\rho_1^2}{(1-\lambda^2)s_1}\sum_{k=1}^{K}\frac{1}{n}\sum_{j=1}^{n}\mathbb{E}[\|\nabla_{\mathbf{x}}f_j(\mathbf{x}_t^{(k)}, \mathbf{y}_t^{(k)}) - \mathbf{g}_{j,t}^{(k)}\|^2] \ .
$$

*Proof.* Based on the gradient tracking scheme in Algorithm 1,we have

$$
\begin{aligned}
\sum_{k=1}^{K} \mathbb{E}[\|\mathbf{a}_{t+1}^{(k)} - \bar{\mathbf{a}}_{t+1}\|^2] &= \mathbb{E}[\|A_{t+1} - \bar{A}_{t+1}\|_F^2] \\
&= \mathbb{E}[\|A_t W + V_{t+1} - V_t - (\bar{A}_t + \bar{V}_{t+1} - \bar{V}_t)\|_F^2] \\
&\leq (1+a)\mathbb{E}[\|A_t W - \bar{A}_t\|_F^2] + (1+\frac{1}{a})\mathbb{E}[\|V_{t+1} - V_t - \bar{V}_{t+1} + \bar{V}_t\|_F^2] \\
&\leq (1+a)\lambda^2 \mathbb{E}[\|A_t - \bar{A}_t\|_F^2] + (1+\frac{1}{a})\mathbb{E}[\|V_{t+1} - V_t\|_F^2] \\
&= \frac{1+\lambda^2}{2}\mathbb{E}[\|A_t - \bar{A}_t\|_F^2] + \frac{2}{1-\lambda^2}\mathbb{E}[\|V_{t+1} - V_t\|_F^2] \\
&= \frac{1+\lambda^2}{2}\sum_{k=1}^{K}\mathbb{E}[\|\mathbf{a}_t^{(k)} - \bar{\mathbf{a}}_t\|^2] + \frac{2}{1-\lambda^2}\sum_{k=1}^{K}\mathbb{E}[\|\mathbf{v}_{t+1}^{(k)} - \mathbf{v}_t^{(k)}\|^2] \;,
\end{aligned}
\tag{23}
$$

where the second to last step holds due to $a = \frac{1-\lambda^2}{2\lambda^2}$. In the following, we bound $\mathbb{E}[\|\mathbf{v}_{t+1}^{(k)} - \mathbf{v}_t^{(k)}\|^2]$.

$$
\begin{aligned}
&\mathbb{E}[\|\mathbf{v}_{t+1}^{(k)} - \mathbf{v}_t^{(k)}\|^2] \\
&= \mathbb{E}[\|\frac{1}{s_{t+1}}\sum_{i\in\mathcal{S}_{t+1}}\left(\nabla_{\mathbf{x}}f_i(\mathbf{x}_{t+1}^{(k)},\mathbf{y}_{t+1}^{(k)}) - \nabla_{\mathbf{x}}f_i(\mathbf{x}_t^{(k)},\mathbf{y}_t^{(k)})\right) + (1-\rho_{t+1})\mathbf{v}_t^{(k)} \\
&\quad + \rho_{t+1}\Big(\frac{1}{s_{t+1}}\sum_{i\in\mathcal{S}_{t+1}}(\nabla_{\mathbf{x}}f_i(\mathbf{x}_t^{(k)},\mathbf{y}_t^{(k)}) - \mathbf{g}_{i,t}^{(k)}) + \frac{1}{n}\sum_{j=1}^{n}\mathbf{g}_{j,t}^{(k)}\Big) - \mathbf{v}_t^{(k)}\|^2] \\
&= \mathbb{E}[\|\frac{1}{s_{t+1}}\sum_{i\in\mathcal{S}_{t+1}}\left(\nabla_{\mathbf{x}}f_i(\mathbf{x}_{t+1}^{(k)},\mathbf{y}_{t+1}^{(k)}) - \nabla_{\mathbf{x}}f_i(\mathbf{x}_t^{(k)},\mathbf{y}_t^{(k)})\right) - \rho_{t+1}\mathbf{v}_t^{(k)} + \rho_{t+1}\nabla_{\mathbf{x}}f(\mathbf{x}_t^{(k)},\mathbf{y}_t^{(k)}) \\
&\quad + \rho_{t+1}\Big(\frac{1}{s_{t+1}}\sum_{i\in\mathcal{S}_{t+1}}(\nabla_{\mathbf{x}}f_i(\mathbf{x}_t^{(k)},\mathbf{y}_t^{(k)}) - \mathbf{g}_{i,t}^{(k)}) + \frac{1}{n}\sum_{j=1}^{n}\mathbf{g}_{j,t}^{(k)} - \nabla_{\mathbf{x}}f(\mathbf{x}_t^{(k)},\mathbf{y}_t^{(k)})\Big)\|^2] \\
&\leq 3\mathbb{E}[\|\frac{1}{s_{t+1}}\sum_{i\in\mathcal{S}_{t+1}}\left(\nabla_{\mathbf{x}}f_i(\mathbf{x}_{t+1}^{(k)},\mathbf{y}_{t+1}^{(k)}) - \nabla_{\mathbf{x}}f_i(\mathbf{x}_t^{(k)},\mathbf{y}_t^{(k)})\right)\|^2] \\
&\quad + 3\mathbb{E}[\| - \rho_{t+1}\mathbf{v}_t^{(k)} + \rho_{t+1}\nabla_{\mathbf{x}}f(\mathbf{x}_t^{(k)},\mathbf{y}_t^{(k)})\|^2] \\
&\quad + 3\mathbb{E}[\|\rho_{t+1}\Big(\frac{1}{s_{t+1}}\sum_{i\in\mathcal{S}_{t+1}}(\nabla_{\mathbf{x}}f_i(\mathbf{x}_t^{(k)},\mathbf{y}_t^{(k)}) - \mathbf{g}_{i,t}^{(k)}) + \frac{1}{n}\sum_{j=1}^{n}\mathbf{g}_{j,t}^{(k)} - \nabla_{\mathbf{x}}f(\mathbf{x}_t^{(k)},\mathbf{y}_t^{(k)})\Big)\|^2] \\
&\leq \frac{3L^2}{s_{t+1}}\mathbb{E}[\|\mathbf{x}_{t+1}^{(k)} - \mathbf{x}_t^{(k)}\|^2] + \frac{3L^2}{s_{t+1}}\mathbb{E}[\|\mathbf{y}_{t+1}^{(k)} - \mathbf{y}_t^{(k)}\|^2] + 3\rho_{t+1}^2\mathbb{E}[\|\mathbf{v}_t^{(k)} - \nabla_{\mathbf{x}}f(\mathbf{x}_t^{(k)},\mathbf{y}_t^{(k)})\|^2] \\
&\quad + \frac{3\rho_{t+1}^2}{s_{t+1}}\frac{1}{n}\sum_{j=1}^{n}\mathbb{E}[\|\nabla_{\mathbf{x}}f_j(\mathbf{x}_t^{(k)},\mathbf{y}_t^{(k)}) - \mathbf{g}_{j,t}^{(k)}\|^2] \;.
\end{aligned}
\tag{24}
$$

Then, by setting $s_t = s_1$, $\rho_t = \rho_1$ and combining above two inequalities, we have

$$
\begin{aligned}
\sum_{k=1}^{K}\mathbb{E}[\|\mathbf{a}_{t+1}^{(k)} - \bar{\mathbf{a}}_{t+1}\|^2] &\leq \frac{1+\lambda^2}{2}\sum_{k=1}^{K}\mathbb{E}[\|\mathbf{a}_t^{(k)} - \bar{\mathbf{a}}_t\|^2] + \frac{6L^2}{(1-\lambda^2)s_1}\sum_{k=1}^{K}\mathbb{E}[\|\mathbf{x}_{t+1}^{(k)} - \mathbf{x}_t^{(k)}\|^2] \\
&\quad + \frac{6L^2}{(1-\lambda^2)s_1}\sum_{k=1}^{K}\mathbb{E}[\|\mathbf{y}_{t+1}^{(k)} - \mathbf{y}_t^{(k)}\|^2] + \frac{6\rho_1^2}{1-\lambda^2}\sum_{k=1}^{K}\mathbb{E}[\|\mathbf{v}_t^{(k)} - \nabla_{\mathbf{x}}f(\mathbf{x}_t^{(k)},\mathbf{y}_t^{(k)})\|^2] \\
&\quad + \frac{6\rho_1^2}{(1-\lambda^2)s_1}\sum_{k=1}^{K}\frac{1}{n}\sum_{j=1}^{n}\mathbb{E}[\|\nabla_{\mathbf{x}}f_j(\mathbf{x}_t^{(k)},\mathbf{y}_t^{(k)}) - \mathbf{g}_{j,t}^{(k)}\|^2] \;.
\end{aligned}
\tag{25}
$$

$\square$

**Lemma 4.** *Given Assumption 1-3, for $t > 0$, by setting $s_t = s_1$, $\rho_t = \rho_1$, we have*

$$
\begin{aligned}
\sum_{k=1}^{K} \mathbb{E}[\|\mathbf{b}_{t+1}^{(k)} - \bar{\mathbf{b}}_{t+1}\|^2] \leq {} & \frac{1+\lambda^2}{2} \sum_{k=1}^{K} \mathbb{E}[\|\mathbf{b}_t^{(k)} - \bar{\mathbf{b}}_t\|^2] \\
& + \frac{6L^2}{(1-\lambda^2)s_1} \sum_{k=1}^{K} \mathbb{E}[\|\mathbf{x}_{t+1}^{(k)} - \mathbf{x}_t^{(k)}\|^2] \\
& + \frac{6L^2}{(1-\lambda^2)s_1} \sum_{k=1}^{K} \mathbb{E}[\|\mathbf{y}_{t+1}^{(k)} - \mathbf{y}_t^{(k)}\|^2] \\
& + \frac{6\rho_1^2}{1-\lambda^2} \sum_{k=1}^{K} \mathbb{E}[\|\mathbf{u}_t^{(k)} - \nabla_\mathbf{y} f(\mathbf{x}_t^{(k)}, \mathbf{y}_t^{(k)})\|^2] \\
& + \frac{6\rho_1^2}{(1-\lambda^2)s_1} \sum_{k=1}^{K} \frac{1}{n} \sum_{j=1}^{n} \mathbb{E}[\|\nabla_\mathbf{y} f_j(\mathbf{x}_t^{(k)}, \mathbf{y}_t^{(k)}) - \mathbf{h}_{j,t}^{(k)}\|^2] .
\end{aligned}
\tag{26}
$$

Similarly, we can prove the inequality with respect to $\mathbb{E}[\|\mathbf{b}_{t+1}^{(k)} - \bar{\mathbf{b}}_{t+1}\|^2]$ by following Lemma 3.

**Lemma 5.** *Given Assumption 1-3, for $t > 0$, by setting $s_t = s_1$, and $\rho_t = \rho_1$, we have*

$$
\begin{aligned}
& \mathbb{E}[\|\mathbf{v}_{t+1}^{(k)} - \nabla_\mathbf{x} f(\mathbf{x}_{t+1}^{(k)}, \mathbf{y}_{t+1}^{(k)})\|^2] \\
\leq {} & \frac{2L^2}{s_1} \mathbb{E}[\|\mathbf{x}_{t+1}^{(k)} - \mathbf{x}_t^{(k)}\|^2] + \frac{2L^2}{s_1} \mathbb{E}[\|\mathbf{y}_{t+1}^{(k)} - \mathbf{y}_t^{(k)}\|^2] \\
& + \frac{2\rho_1^2}{s_1} \frac{1}{n} \sum_{i=1}^{n} \mathbb{E}[\|\nabla_\mathbf{x} f_i(\mathbf{x}_t^{(k)}, \mathbf{y}_t^{(k)}) - \mathbf{g}_{i,t}^{(k)}\|^2] \\
& + (1-\rho_1)^2 \mathbb{E}[\|\mathbf{v}_t^{(k)} - \nabla_\mathbf{x} f(\mathbf{x}_t^{(k)}, \mathbf{y}_t^{(k)})\|^2] .
\end{aligned}
\tag{27}
$$

*When $t = 0$, $\rho_0 = 1$, we have*

$$
\begin{aligned}
& \mathbb{E}[\|\mathbf{v}_0 - \nabla_\mathbf{x} f(\mathbf{x}_0^{(k)}, \mathbf{y}_0^{(k)})\|^2] \\
= {} & \frac{n - s_0}{(n-1)s_0} \frac{1}{n} \sum_{i=1}^{n} \|\nabla_\mathbf{x} f_i(\mathbf{x}_0^{(k)}, \mathbf{y}_0^{(k)})\|^2 .
\end{aligned}
\tag{28}
$$

*Proof.* When $t = 0$, $\rho_0 = 1$, we can get

$$
\begin{aligned}
& \mathbb{E}[\|\mathbf{v}_0 - \nabla_\mathbf{x} f(\mathbf{x}_0^{(k)}, \mathbf{y}_0^{(k)})\|^2] \\
= {} & \mathbb{E}[\|\frac{1}{s_0} \sum_{i \in \mathcal{S}_0} (\nabla_\mathbf{x} f_i(\mathbf{x}_0^{(k)}, \mathbf{y}_0^{(k)}) - \nabla_\mathbf{x} f(\mathbf{x}_0^{(k)}, \mathbf{y}_0^{(k)}))\|^2] \\
= {} & \frac{n - s_0}{(n-1)s_0} \frac{1}{n} \sum_{i=1}^{n} \|\nabla_\mathbf{x} f_i(\mathbf{x}_0^{(k)}, \mathbf{y}_0^{(k)})\|^2 .
\end{aligned}
\tag{29}
$$

For $t > 0$, we set the batch size to $s_t = s_1$ and $\rho_t = \rho_1$. We can get

$$
\begin{aligned}
&\mathbb{E}[\|\mathbf{v}_{t+1}^{(k)} - \nabla_{\mathbf{x}}f(\mathbf{x}_{t+1}^{(k)}, \mathbf{y}_{t+1}^{(k)})\|^2] \\
&= \mathbb{E}[\|\frac{1}{s_{t+1}} \sum_{i \in \mathcal{S}_{t+1}} \Big(\nabla_{\mathbf{x}}f_i(\mathbf{x}_{t+1}^{(k)}, \mathbf{y}_{t+1}^{(k)}) - \nabla_{\mathbf{x}}f_i(\mathbf{x}_t^{(k)}, \mathbf{y}_t^{(k)})\Big) + (1 - \rho_{t+1})\mathbf{v}_t^{(k)} \\
&\quad + \rho_{t+1}\Big(\frac{1}{s_{t+1}} \sum_{i \in \mathcal{S}_{t+1}} (\nabla_{\mathbf{x}}f_i(\mathbf{x}_t^{(k)}, \mathbf{y}_t^{(k)}) - \mathbf{g}_{i,t}^{(k)}) + \frac{1}{n} \sum_{j=1}^n \mathbf{g}_{j,t}^{(k)}\Big) - \nabla_{\mathbf{x}}f(\mathbf{x}_{t+1}^{(k)}, \mathbf{y}_{t+1}^{(k)})\|^2] \\
&= \mathbb{E}[\|\frac{1}{s_{t+1}} \sum_{i \in \mathcal{S}_{t+1}} \Big(\nabla_{\mathbf{x}}f_i(\mathbf{x}_{t+1}^{(k)}, \mathbf{y}_{t+1}^{(k)}) - \nabla_{\mathbf{x}}f_i(\mathbf{x}_t^{(k)}, \mathbf{y}_t^{(k)}) - \nabla_{\mathbf{x}}f(\mathbf{x}_{t+1}^{(k)}, \mathbf{y}_{t+1}^{(k)}) + \nabla_{\mathbf{x}}f(\mathbf{x}_t^{(k)}, \mathbf{y}_t^{(k)})\Big) \\
&\quad + \rho_{t+1}\Big(\frac{1}{s_{t+1}} \sum_{i \in \mathcal{S}_{t+1}} (\nabla_{\mathbf{x}}f_i(\mathbf{x}_t^{(k)}, \mathbf{y}_t^{(k)}) - \mathbf{g}_{i,t}^{(k)}) + \frac{1}{n} \sum_{j=1}^n \mathbf{g}_{j,t}^{(k)} - \nabla_{\mathbf{x}}f(\mathbf{x}_t^{(k)}, \mathbf{y}_t^{(k)})\Big) \\
&\quad + (1 - \rho_{t+1})(\mathbf{v}_t^{(k)} - \nabla_{\mathbf{x}}f(\mathbf{x}_t^{(k)}, \mathbf{y}_t^{(k)}))\|^2] \\
&\leq \mathbb{E}[\|\frac{1}{s_{t+1}} \sum_{i \in \mathcal{S}_{t+1}} \Big(\nabla_{\mathbf{x}}f_i(\mathbf{x}_{t+1}^{(k)}, \mathbf{y}_{t+1}^{(k)}) - \nabla_{\mathbf{x}}f_i(\mathbf{x}_t^{(k)}, \mathbf{y}_t^{(k)}) - \nabla_{\mathbf{x}}f(\mathbf{x}_{t+1}^{(k)}, \mathbf{y}_{t+1}^{(k)}) + \nabla_{\mathbf{x}}f(\mathbf{x}_t^{(k)}, \mathbf{y}_t^{(k)})\Big) \\
&\quad + \rho_{t+1}\Big(\frac{1}{s_{t+1}} \sum_{i \in \mathcal{S}_{t+1}} (\nabla_{\mathbf{x}}f_i(\mathbf{x}_t^{(k)}, \mathbf{y}_t^{(k)}) - \mathbf{g}_{i,t}^{(k)}) + \frac{1}{n} \sum_{j=1}^n \mathbf{g}_{j,t}^{(k)} - \nabla_{\mathbf{x}}f(\mathbf{x}_t^{(k)}, \mathbf{y}_t^{(k)})\Big)\|^2] \\
&\quad + (1 - \rho_{t+1})^2 \mathbb{E}[\|\mathbf{v}_t^{(k)} - \nabla_{\mathbf{x}}f(\mathbf{x}_t^{(k)}, \mathbf{y}_t^{(k)})\|^2] \\
&\leq 2\mathbb{E}[\|\frac{1}{s_{t+1}} \sum_{i \in \mathcal{S}_{t+1}} \Big(\nabla_{\mathbf{x}}f_i(\mathbf{x}_{t+1}^{(k)}, \mathbf{y}_{t+1}^{(k)}) - \nabla_{\mathbf{x}}f_i(\mathbf{x}_t^{(k)}, \mathbf{y}_t^{(k)}) - \nabla_{\mathbf{x}}f(\mathbf{x}_{t+1}^{(k)}, \mathbf{y}_{t+1}^{(k)}) + \nabla_{\mathbf{x}}f(\mathbf{x}_t^{(k)}, \mathbf{y}_t^{(k)})\Big)\|^2] \quad (30) \\
&\quad + 2\rho_{t+1}^2 \mathbb{E}[\|\Big(\frac{1}{s_{t+1}} \sum_{i \in \mathcal{S}_{t+1}} (\nabla_{\mathbf{x}}f_i(\mathbf{x}_t^{(k)}, \mathbf{y}_t^{(k)}) - \mathbf{g}_{i,t}^{(k)}) + \frac{1}{n} \sum_{j=1}^n \mathbf{g}_{j,t}^{(k)} - \nabla_{\mathbf{x}}f(\mathbf{x}_t^{(k)}, \mathbf{y}_t^{(k)})\Big)\|^2] \\
&\quad + (1 - \rho_{t+1})^2 \mathbb{E}[\|\mathbf{v}_t^{(k)} - \nabla_{\mathbf{x}}f(\mathbf{x}_t^{(k)}, \mathbf{y}_t^{(k)})\|^2] \\
&\leq \frac{2}{s_{t+1}^2} \sum_{i \in \mathcal{S}_{t+1}} \mathbb{E}[\|\nabla_{\mathbf{x}}f_i(\mathbf{x}_{t+1}^{(k)}, \mathbf{y}_{t+1}^{(k)}) - \nabla_{\mathbf{x}}f_i(\mathbf{x}_t^{(k)}, \mathbf{y}_t^{(k)}) - \nabla_{\mathbf{x}}f(\mathbf{x}_{t+1}^{(k)}, \mathbf{y}_{t+1}^{(k)}) + \nabla_{\mathbf{x}}f(\mathbf{x}_t^{(k)}, \mathbf{y}_t^{(k)})\|^2] \\
&\quad + \frac{2\rho_{t+1}^2}{s_{t+1}^2} \sum_{i \in \mathcal{S}_{t+1}} \mathbb{E}[\|\nabla_{\mathbf{x}}f_i(\mathbf{x}_t^{(k)}, \mathbf{y}_t^{(k)}) - \mathbf{g}_{i,t}^{(k)} + \frac{1}{n} \sum_{j=1}^n \mathbf{g}_{j,t}^{(k)} - \nabla_{\mathbf{x}}f(\mathbf{x}_t^{(k)}, \mathbf{y}_t^{(k)})\|^2] \\
&\quad + (1 - \rho_{t+1})^2 \mathbb{E}[\|(\mathbf{v}_t^{(k)} - \nabla_{\mathbf{x}}f(\mathbf{x}_t^{(k)}, \mathbf{y}_t^{(k)}))\|^2] \\
&\leq \frac{2}{s_{t+1}^2} \sum_{i \in \mathcal{S}_{t+1}} \mathbb{E}[\|\nabla_{\mathbf{x}}f_i(\mathbf{x}_{t+1}^{(k)}, \mathbf{y}_{t+1}^{(k)}) - \nabla_{\mathbf{x}}f_i(\mathbf{x}_t^{(k)}, \mathbf{y}_t^{(k)})\|^2] + \frac{2\rho_{t+1}^2}{s_{t+1}^2} \sum_{i \in \mathcal{S}_{t+1}} \mathbb{E}[\|\nabla_{\mathbf{x}}f_i(\mathbf{x}_t^{(k)}, \mathbf{y}_t^{(k)}) - \mathbf{g}_{i,t}^{(k)}\|^2] \\
&\quad + (1 - \rho_{t+1})^2 \mathbb{E}[\|(\mathbf{v}_t^{(k)} - \nabla_{\mathbf{x}}f(\mathbf{x}_t^{(k)}, \mathbf{y}_t^{(k)}))\|^2] \\
&\leq \frac{2L^2}{s_{t+1}} \mathbb{E}[\|\mathbf{x}_{t+1}^{(k)} - \mathbf{x}_t^{(k)}\|^2] + \frac{2L^2}{s_{t+1}} \mathbb{E}[\|\mathbf{y}_{t+1}^{(k)} - \mathbf{y}_t^{(k)}\|^2] + \frac{2\rho_{t+1}^2}{s_{t+1}} \frac{1}{n} \sum_{j=1}^n \mathbb{E}[\|\nabla_{\mathbf{x}}f_j(\mathbf{x}_t^{(k)}, \mathbf{y}_t^{(k)}) - \mathbf{g}_{j,t}^{(k)}\|^2] \\
&\quad + (1 - \rho_{t+1})^2 \mathbb{E}[\|\mathbf{v}_t^{(k)} - \nabla_{\mathbf{x}}f(\mathbf{x}_t^{(k)}, \mathbf{y}_t^{(k)})\|^2] ,
\end{aligned}
$$

where the second to last step follows from Eq. (31), the last step follows from Assumption 2.

$$
\begin{aligned}
&\mathbb{E}[\|\nabla_{\mathbf{x}}f_i(\mathbf{x}_t^{(k)}, \mathbf{y}_t^{(k)}) - \mathbf{g}_{i,t}^{(k)} + \frac{1}{n} \sum_{j=1}^n \mathbf{g}_{j,t}^{(k)} - \nabla_{\mathbf{x}}f(\mathbf{x}_t^{(k)}, \mathbf{y}_t^{(k)})\|^2] \leq \mathbb{E}[\|\nabla_{\mathbf{x}}f_i(\mathbf{x}_t^{(k)}, \mathbf{y}_t^{(k)}) - \mathbf{g}_{i,t}^{(k)}\|^2] \\
&= \mathbb{E}[\|\sum_{j=1}^n \mathbf{1}_{\{i==j\}} \nabla_{\mathbf{x}}f_j(\mathbf{x}_t^{(k)}, \mathbf{y}_t^{(k)}) - \mathbf{g}_{j,t}^{(k)}\|^2] = \frac{1}{n} \sum_{j=1}^n \mathbb{E}[\|\nabla_{\mathbf{x}}f_j(\mathbf{x}_t^{(k)}, \mathbf{y}_t^{(k)}) - \mathbf{g}_{j,t}^{(k)}\|^2] .
\end{aligned}
$$

$$(31)$$

□

**Lemma 6.** *Given Assumption 1-3, for $t > 0$, by setting $s_t = s_1$ and $\rho_t = \rho_1$, we have*

$$
\mathbb{E}[\|\mathbf{u}_{t+1}^{(k)} - \nabla_{\mathbf{y}} f(\mathbf{x}_{t+1}^{(k)}, \mathbf{y}_{t+1}^{(k)})\|^2]
$$
$$
\leq \frac{2L^2}{s_1} \mathbb{E}[\|\mathbf{x}_{t+1}^{(k)} - \mathbf{x}_t^{(k)}\|^2] + \frac{2L^2}{s_1} \mathbb{E}[\|\mathbf{y}_{t+1}^{(k)} - \mathbf{y}_t^{(k)}\|^2]
$$
$$
+ \frac{2\rho_1^2}{s_1} \frac{1}{n} \sum_{i=1}^n \mathbb{E}[\|\nabla_{\mathbf{y}} f_i(\mathbf{x}_t^{(k)}, \mathbf{y}_t^{(k)}) - \mathbf{h}_{i,t}^{(k)}\|^2]
$$
$$
+ (1 - \rho_1)^2 \mathbb{E}[\|\mathbf{u}_t^{(k)} - \nabla_{\mathbf{y}} f(\mathbf{x}_t^{(k)}, \mathbf{y}_t^{(k)})\|^2] \,.
$$
(32)

*For $t = 0$, $\rho_0 = 1$, we have*

$$
\mathbb{E}[\|\mathbf{u}_0^{(k)} - \nabla_{\mathbf{y}} f(\mathbf{x}_0^{(k)}, \mathbf{y}_0^{(k)})\|^2]
$$
$$
= \frac{n - s_0}{(n-1)s_0} \frac{1}{n} \sum_{i=1}^n \|\nabla_{\mathbf{y}} f_i(\mathbf{x}_0^{(k)}, \mathbf{y}_0^{(k)})\|^2 \,.
$$
(33)

Lemma 6 can be proved by following Lemma 5. Thus, we ignore it.

**Lemma 7.** *Given Assumption 1-3, for $t > 0$, by setting $s_t = s_1$ and $\alpha_t = \alpha_1$, we have*

$$
\mathbb{E}[\frac{1}{n} \sum_{j=1}^n \|\nabla_{\mathbf{x}} f_j(\mathbf{x}_{t+1}^{(k)}, \mathbf{y}_{t+1}^{(k)}) - \mathbf{g}_{j,t+1}^{(k)}\|^2]
$$
$$
\leq 2L^2 (1 - \frac{s_1}{n})(1 + \frac{1}{\alpha_1}) \|\mathbf{x}_{t+1}^{(k)} - \mathbf{x}_t^{(k)}\|^2
$$
$$
+ 2L^2 (1 - \frac{s_1}{n})(1 + \frac{1}{\alpha_1}) \|\mathbf{y}_{t+1}^{(k)} - \mathbf{y}_t^{(k)}\|^2
$$
$$
+ (1 - \frac{s_1}{n})(1 + \alpha_1) \frac{1}{n} \sum_{j=1}^n \|\nabla_{\mathbf{x}} f_j(\mathbf{x}_t^{(k)}, \mathbf{y}_t^{(k)}) - \mathbf{g}_{j,t}^{(k)}\|^2 \,.
$$
(34)

*For $t = 0$, we have*

$$
\mathbb{E}[\frac{1}{n} \sum_{j=1}^n \|\nabla_{\mathbf{x}} f_j(\mathbf{x}_0^{(k)}, \mathbf{y}_0^{(k)}) - \mathbf{g}_{j,0}^{(k)}\|^2]
$$
$$
= (1 - \frac{s_0}{n}) \frac{1}{n} \sum_{j=1}^n \|\nabla_{\mathbf{x}} f_j(\mathbf{x}_0^{(k)}, \mathbf{y}_0^{(k)})\|^2 \,.
$$
(35)

*Proof.* For $t > 0$, we have

$$
\mathbb{E}[\frac{1}{n} \sum_{j=1}^n \|\nabla_{\mathbf{x}} f_j(\mathbf{x}_{t+1}^{(k)}, \mathbf{y}_{t+1}^{(k)}) - \mathbf{g}_{j,t+1}^{(k)}\|^2]
$$
$$
= (1 - \frac{s_{t+1}}{n}) \frac{1}{n} \sum_{j=1}^n \mathbb{E}[\|\nabla_{\mathbf{x}} f_j(\mathbf{x}_{t+1}^{(k)}, \mathbf{y}_{t+1}^{(k)}) - \mathbf{g}_{j,t}^{(k)}\|^2]
$$
$$
= (1 - \frac{s_{t+1}}{n}) \frac{1}{n} \sum_{j=1}^n \mathbb{E}[\|\nabla_{\mathbf{x}} f_j(\mathbf{x}_{t+1}^{(k)}, \mathbf{y}_{t+1}^{(k)}) - \nabla_{\mathbf{x}} f_j(\mathbf{x}_t^{(k)}, \mathbf{y}_t^{(k)}) + \nabla_{\mathbf{x}} f_j(\mathbf{x}_t^{(k)}, \mathbf{y}_t^{(k)}) - \mathbf{g}_{j,t}^{(k)}\|^2]
$$
$$
\leq 2L^2 (1 - \frac{s_{t+1}}{n})(1 + \frac{1}{\alpha_{t+1}}) \mathbb{E}[\|\mathbf{x}_{t+1}^{(k)} - \mathbf{x}_t^{(k)}\|^2] + 2L^2 (1 - \frac{s_{t+1}}{n})(1 + \frac{1}{\alpha_{t+1}}) \mathbb{E}[\|\mathbf{y}_{t+1}^{(k)} - \mathbf{y}_t^{(k)}\|^2]
$$
$$
+ (1 - \frac{s_{t+1}}{n})(1 + \alpha_{t+1}) \frac{1}{n} \sum_{j=1}^n \mathbb{E}[\|\nabla_{\mathbf{x}} f_j(\mathbf{x}_t^{(k)}, \mathbf{y}_t^{(k)}) - \mathbf{g}_{j,t}^{(k)}\|^2] \,,
$$
(36)

where $\alpha_{t+1} > 0$. By setting $s_t = s_1$ and $\alpha_t = \alpha_1$, we complete the proof for the first part.

For $t = 0$, we have

$$\mathbb{E}[\frac{1}{n}\sum_{j=1}^{n}\|\nabla_{\mathbf{x}}f_j(\mathbf{x}_0^{(k)}, \mathbf{y}_0^{(k)}) - \mathbf{g}_{j,0}^{(k)}\|^2] = (1 - \frac{s_0}{n})\frac{1}{n}\sum_{j=1}^{n}\|\nabla_{\mathbf{x}}f_j(\mathbf{x}_0^{(k)}, \mathbf{y}_0^{(k)})\|^2 \ . \tag{37}$$

$\square$

**Lemma 8.** *Given Assumption 1-3, for $t > 0$, by setting $s_t = s_1$ and $\alpha_t = \alpha_1$, we have*

$$\mathbb{E}[\frac{1}{n}\sum_{j=1}^{n}\|\nabla_{\mathbf{y}}f_j(\mathbf{x}_{t+1}^{(k)}, \mathbf{y}_{t+1}^{(k)}) - \mathbf{h}_{j,t+1}^{(k)}\|^2]$$

$$\leq 2L^2(1 - \frac{s_1}{n})(1 + \frac{1}{\alpha_t})\|\mathbf{x}_{t+1}^{(k)} - \mathbf{x}_t^{(k)}\|^2$$

$$+ 2L^2(1 - \frac{s_1}{n})(1 + \frac{1}{\alpha_t})\|\mathbf{y}_{t+1}^{(k)} - \mathbf{y}_t^{(k)}\|^2 \tag{38}$$

$$+ (1 - \frac{s_1}{n})(1 + \alpha_t)\frac{1}{n}\sum_{j=1}^{n}\|\nabla_{\mathbf{y}}f_j(\mathbf{x}_t^{(k)}, \mathbf{y}_t^{(k)}) - \mathbf{h}_{j,t}^{(k)}\|^2 \ .$$

*For $t = 0$, we have*

$$\mathbb{E}[\frac{1}{n}\sum_{j=1}^{n}\|\nabla_{\mathbf{y}}f_j(\mathbf{x}_0^{(k)}, \mathbf{y}_0^{(k)}) - \mathbf{h}_{j,0}\|^2]$$

$$= (1 - \frac{s_0}{n})\frac{1}{n}\sum_{j=1}^{n}\|\nabla_{\mathbf{y}}f_j(\mathbf{x}_0^{(k)}, \mathbf{y}_0^{(k)})\|^2 \ . \tag{39}$$

Lemma 8 can be proved by following Lemma 7. Thus, we do not include it.

**Lemma 9.** *Given Assumption 1-3, we have*

$$\sum_{k=1}^{K}\|\mathbf{x}_{t+1}^{(k)} - \bar{\mathbf{x}}_{t+1}\|^2 \leq \Big(1 - \frac{\eta(1 - \lambda^2)}{2}\Big)\sum_{k=1}^{K}\|\mathbf{x}_t^{(k)} - \bar{\mathbf{x}}_t\|^2 + \frac{2\eta\gamma_1^2}{1 - \lambda^2}\sum_{k=1}^{K}\|\mathbf{a}_t^{(k)} - \bar{\mathbf{a}}_t\|^2 \ . \tag{40}$$

*Proof.*

$$\sum_{k=1}^{K}\|\mathbf{x}_{t+1}^{(k)} - \bar{\mathbf{x}}_{t+1}\|^2 = \|X_{t+1} - \bar{X}_{t+1}\|_F^2$$

$$= \|(1 - \eta)X_t + \eta X_{t+\frac{1}{2}} - (1 - \eta)\bar{X}_t - \eta\bar{X}_{t+\frac{1}{2}}\|_F^2$$

$$\leq (1 + a_0)(1 - \eta)^2\|X_t - \bar{X}_t\|_F^2 + \eta^2(1 + \frac{1}{a_0})\|X_{t+\frac{1}{2}} - \bar{X}_{t+\frac{1}{2}}\|_F^2$$

$$\leq (1 - \eta)\|X_t - \bar{X}_t\|_F^2 + \eta\|X_{t+\frac{1}{2}} - \bar{X}_{t+\frac{1}{2}}\|_F^2$$

$$\leq (1 - \eta)\|X_t - \bar{X}_t\|_F^2 + \eta\|X_t W + \gamma_1 A_t - (\bar{X}_t + \gamma_1\bar{A}_t)\|_F^2 \tag{41}$$

$$\leq (1 - \eta)\|X_t - \bar{X}_t\|_F^2 + \eta(1 + a_1)\|X_t W - \bar{X}_t\|_F^2 + \eta\gamma_1^2(1 + \frac{1}{a_1})\|A_t - \bar{A}_t\|_F^2$$

$$\leq (1 - \eta)\|X_t - \bar{X}_t\|_F^2 + \frac{\eta(1 + \lambda^2)}{2}\|X_t - \bar{X}_t\|_F^2 + \frac{2\eta\gamma_1^2}{1 - \lambda^2}\|A_t - \bar{A}_t\|_F^2$$

$$= \Big(1 - \eta + \frac{\eta(1 + \lambda^2)}{2}\Big)\|X_t - \bar{X}_t\|_F^2 + \frac{2\eta\gamma_1^2}{1 - \lambda^2}\|A_t - \bar{A}_t\|_F^2$$

$$= \Big(1 - \frac{\eta(1 - \lambda^2)}{2}\Big)\sum_{k=1}^{K}\|\mathbf{x}_t^{(k)} - \bar{\mathbf{x}}_t\|^2 + \frac{2\eta\gamma_1^2}{1 - \lambda^2}\sum_{k=1}^{K}\|\mathbf{a}_t^{(k)} - \bar{\mathbf{a}}_t\|^2 \ ,$$

where the second inequality follows from $a_0 = \frac{\eta}{1-\eta}$, the last inequality follows from $a_1 = \frac{1-\lambda^2}{2\lambda^2}$ and $\|X_t W - \bar{X}_t\|_F^2 \leq \lambda^2 \|X_t - \bar{X}_t\|_F^2$.

$\square$

**Lemma 10.** *Given Assumption 1-3, we have*

$$\sum_{k=1}^{K} \|\mathbf{y}_{t+1}^{(k)} - \bar{\mathbf{y}}_{t+1}\|^2 \leq \left(1 - \frac{\eta(1-\lambda^2)}{2}\right) \sum_{k=1}^{K} \|\mathbf{y}_t^{(k)} - \bar{\mathbf{y}}_t\|^2 + \frac{2\eta\gamma_2^2}{1-\lambda^2} \sum_{k=1}^{K} \|\mathbf{b}_t^{(k)} - \bar{\mathbf{b}}_t\|^2 \ . \tag{42}$$

Similarly, we can prove the second inequality regarding $\mathbf{y}$ in Lemma 9.

**Lemma 11.** *Given Assumption 1-3, we have*

$$\|X_{t+1} - X_t\|_F^2 \leq 12\eta^2 \sum_{k=1}^{K} \|\mathbf{x}_t^{(k)} - \bar{\mathbf{x}}_t\|^2 + 3\gamma_1^2\eta^2 \sum_{k=1}^{K} \|\mathbf{a}_t^{(k)} - \bar{\mathbf{a}}_t\|^2 + 3\gamma_1^2\eta^2 K \|\bar{\mathbf{v}}_t\|^2 \ , \tag{43}$$

$$\|Y_{t+1} - Y_t\|_F^2 \leq 12\eta^2 \sum_{k=1}^{K} \|\mathbf{y}_t^{(k)} - \bar{\mathbf{y}}_t\|^2 + 3\gamma_2^2\eta^2 \sum_{k=1}^{K} \|\mathbf{b}_t^{(k)} - \bar{\mathbf{b}}_t\|^2 + 3\gamma_2^2\eta^2 K \|\bar{\mathbf{u}}_t\|^2 \ . \tag{44}$$

*Proof.*

$$
\begin{aligned}
&\|X_{t+1} - X_t\|_F^2 \\
&= \eta^2 \|X_{t+\frac{1}{2}} - X_t\|_F^2 \\
&= \eta^2 \|X_t W - \gamma_1 A_t - X_t\|_F^2 \\
&= \eta^2 \|X_t W - X_t - \gamma_1 A_t + \gamma_1 \bar{A}_t - \gamma_1 \bar{A}_t\|_F^2 \\
&\leq 3\eta^2 \|X_t W - X_t\|_F^2 + 3\gamma_1^2\eta^2 \|A_t - \bar{A}_t\|_F^2 + 3\gamma_1^2\eta^2 \|\bar{A}_t\|_F^2 \\
&= 3\eta^2 \|(X_t - \bar{X}_t)(W - I)\|_F^2 + 3\gamma_1^2\eta^2 \|A_t - \bar{A}_t\|_F^2 + 3\gamma_1^2\eta^2 \|\bar{A}_t\|_F^2 \\
&\leq 12\eta^2 \|X_t - \bar{X}_t\|_F^2 + 3\gamma_1^2\eta^2 \|A_t - \bar{A}_t\|_F^2 + 3\gamma_1^2\eta^2 \|\bar{V}_t\|_F^2 \\
&= 12\eta^2 \sum_{k=1}^{K} \|\mathbf{x}_t^{(k)} - \bar{\mathbf{x}}_t\|^2 + 3\gamma_1^2\eta^2 \sum_{k=1}^{K} \|\mathbf{a}_t^{(k)} - \bar{\mathbf{a}}_t\|^2 + 3\gamma_1^2\eta^2 K \|\bar{\mathbf{v}}_t\|^2 \ ,
\end{aligned} \tag{45}
$$

where the last inequality follows from $\frac{1}{K} \sum_{k=1}^{K} \mathbf{a}_t^{(k)} = \frac{1}{K} \sum_{k=1}^{K} \mathbf{v}_t^{(k)}$. Similarly, we can prove the inequality for $\|Y_{t+1} - Y_t\|_F^2$.

$\square$

Based on these lemmas, we prove Theorem 1 in the following.

*Proof.* We define the potential function

$$H_t = \mathbb{E}[\Phi(\bar{\mathbf{x}}_t)] + C_0 \mathbb{E}[\|\bar{\mathbf{y}}_t - \mathbf{y}^*(\bar{\mathbf{x}}_t)\|^2]$$

$$+ \frac{C_1}{K} \sum_{k=1}^{K} \mathbb{E}[\|\nabla_{\mathbf{x}} f(\mathbf{x}_t^{(k)}, \mathbf{y}_t^{(k)}) - \mathbf{v}_t^{(k)}\|^2] + \frac{C_2}{K} \sum_{k=1}^{K} \mathbb{E}[\|\nabla_{\mathbf{y}} f(\mathbf{x}_t^{(k)}, \mathbf{y}_t^{(k)}) - \mathbf{u}_t^{(k)}\|^2]$$

$$+ \frac{C_3}{K} \sum_{k=1}^{K} \mathbb{E}[\frac{1}{n} \sum_{j=1}^{n} \|\nabla_{\mathbf{x}} f_j(\mathbf{x}_t^{(k)}, \mathbf{y}_t^{(k)}) - \mathbf{g}_{j,t}^{(k)}\|^2] + \frac{C_4}{K} \sum_{k=1}^{K} \mathbb{E}[\frac{1}{n} \sum_{j=1}^{n} \|\nabla_{\mathbf{y}} f_j(\mathbf{x}_t^{(k)}, \mathbf{y}_t^{(k)}) - \mathbf{h}_{j,t}^{(k)}\|^2] \tag{46}$$

$$+ \frac{C_5}{K} \sum_{k=1}^{K} \mathbb{E}[\|\bar{\mathbf{x}}_t - \mathbf{x}_t^{(k)}\|^2] + \frac{C_6}{K} \sum_{k=1}^{K} \mathbb{E}[\|\bar{\mathbf{y}}_t - \mathbf{y}_t^{(k)}\|^2] + \frac{C_7}{K} \sum_{k=1}^{K} \mathbb{E}[\|\bar{\mathbf{a}}_t - \mathbf{a}_t^{(k)}\|^2] + \frac{C_8}{K} \sum_{k=1}^{K} \mathbb{E}[\|\bar{\mathbf{b}}_t - \mathbf{b}_t^{(k)}\|^2] \ ,$$

where $C_0 = \frac{6\gamma_1 L^2}{\gamma_2 \mu}$, $C_1 = \frac{3\gamma_1\eta}{\rho_1}$, $C_2 = \frac{250\eta\gamma_1 L^2}{\rho_1\mu^2}$, $C_3 = \frac{14n\rho_1\gamma_1\eta}{s_1^2}$, $C_4 = \frac{1012n\rho_1\eta\gamma_1 L^2}{s_1^2\mu^2}$, $C_5 = \frac{106\gamma_1\kappa^2}{(1-\lambda^2)}$, $C_6 = \frac{106\gamma_1\kappa^2 L^2}{(1-\lambda^2)}$, $C_7 = \frac{(1-\lambda^2)\gamma_1\eta}{6\rho_1}$, and $C_8 = \frac{(1-\lambda^2)\eta\gamma_1 L^2}{\rho_1\mu^2}$.

Then, according to aforementioned lemmas, it is easy to get

$$
\begin{aligned}
H_{t+1} - H_t \leq & -\frac{\gamma_1\eta}{2}\mathbb{E}[\|\nabla\Phi(\bar{\mathbf{x}}_t)\|^2] + \Big(\gamma_1\eta L^2 - \frac{\eta\gamma_2\mu}{4}C_0\Big)\mathbb{E}[\|\bar{\mathbf{y}}_t - \mathbf{y}^*(\bar{\mathbf{x}}_t)\|^2] \\
& + \Big(+\frac{25\eta\gamma_1^2\kappa^2}{6\gamma_2\mu}C_0 - \frac{\gamma_1\eta}{4}\Big)\mathbb{E}[\|\bar{\mathbf{v}}_t\|^2] + \Big(-\frac{3\gamma_2^2\eta}{4}C_0\Big)\mathbb{E}[\|\bar{\mathbf{u}}_t\|^2] \\
& + \Big(2\gamma_1\eta L^2 + \frac{25\eta\gamma_2 L^2}{3\mu}C_0 - \frac{\eta(1-\lambda^2)}{2}C_5\Big)\frac{1}{K}\sum_{k=1}^{K}\mathbb{E}[\|\bar{\mathbf{x}}_t - \mathbf{x}_t^{(k)}\|^2] \\
& + \Big(2\gamma_1\eta L^2 + \frac{25\eta\gamma_2 L^2}{3\mu}C_0 - \frac{\eta(1-\lambda^2)}{2}C_6\Big)\frac{1}{K}\sum_{k=1}^{K}\mathbb{E}[\|\bar{\mathbf{y}}_t - \mathbf{y}_t^{(k)}\|^2] \\
& + \Big(2\gamma_1\eta + C_1(1-\rho_1)^2 + \frac{6\rho_1^2}{1-\lambda^2}C_7 - C_1\Big)\frac{1}{K}\sum_{k=1}^{K}\mathbb{E}[\|\nabla_{\mathbf{x}}f(\mathbf{x}_t^{(k)},\mathbf{y}_t^{(k)}) - \mathbf{v}_t^{(k)}\|^2] \\
& + \Big(\frac{25\eta\gamma_2}{3\mu}C_0 + C_2(1-\rho_1)^2 + \frac{6\rho_1^2}{1-\lambda^2}C_8 - C_2\Big)\frac{1}{K}\sum_{k=1}^{K}\mathbb{E}[\|\nabla_{\mathbf{y}}f^{(k)}(\mathbf{x}_t^{(k)},\mathbf{y}_t^{(k)}) - \mathbf{u}_t^{(k)}\|^2] \\
& + \Big(C_3(1-\frac{s_1}{n})(1+\alpha_1) + \frac{2\rho_1^2}{s_1}C_1 + \frac{6\rho_1^2}{(1-\lambda^2)s_1}C_7 - C_3\Big)\frac{1}{K}\sum_{k=1}^{K}\frac{1}{n}\sum_{i=1}^{n}\mathbb{E}[\|\nabla_{\mathbf{x}}f_i(\mathbf{x}_t^{(k)},\mathbf{y}_t^{(k)}) - \mathbf{g}_{i,t}^{(k)}\|^2] \\
& + \Big(C_4(1-\frac{s_1}{n})(1+\alpha_1) + \frac{2\rho_1^2}{s_1}C_2 + \frac{6\rho_1^2}{(1-\lambda^2)s_1}C_8 - C_4\Big)\frac{1}{K}\sum_{k=1}^{K}\frac{1}{n}\sum_{i=1}^{n}\mathbb{E}[\|\nabla_{\mathbf{y}}f_i(\mathbf{x}_t^{(k)},\mathbf{y}_t^{(k)}) - \mathbf{h}_{i,t}^{(k)}\|^2] \\
& + \Big(\frac{4L^2}{s_1}C_1 + \frac{4L^2}{s_1}C_2 + 2L^2(1-\frac{s_1}{n})(1+\frac{1}{\alpha_1})C_3 + 2L^2(1-\frac{s_1}{n})(1+\frac{1}{\alpha_1})C_4 + \frac{6L^2}{(1-\lambda^2)s_1}C_7 \\
& \quad + \frac{6L^2}{(1-\lambda^2)s_1}C_8\Big)\frac{1}{K}\sum_{k=1}^{K}\mathbb{E}[\|\mathbf{x}_{t+1}^{(k)} - \mathbf{x}_t^{(k)}\|^2] \\
& + \Big(\frac{4L^2}{s_1}C_1 + \frac{4L^2}{s_1}C_2 + 2L^2(1-\frac{s_1}{n})(1+\frac{1}{\alpha_1})C_3 + 2L^2(1-\frac{s_1}{n})(1+\frac{1}{\alpha_1})C_4 + \frac{6L^2}{(1-\lambda^2)s_1}C_7 \\
& \quad + \frac{6L^2}{(1-\lambda^2)s_1}C_8\Big)\frac{1}{K}\sum_{k=1}^{K}\mathbb{E}[\|\mathbf{y}_{t+1}^{(k)} - \mathbf{y}_t^{(k)}\|^2] \\
& + \Big(\frac{2\gamma_1^2\eta}{1-\lambda^2}C_5 - \frac{1-\lambda^2}{2}C_7\Big)\frac{1}{K}\sum_{k=1}^{K}\mathbb{E}[\|\mathbf{a}_t^{(k)} - \bar{\mathbf{a}}_t\|^2] + \Big(\frac{2\eta\gamma_2^2}{1-\lambda^2}C_6 - \frac{1-\lambda^2}{2}C_8\Big)\frac{1}{K}\sum_{k=1}^{K}\mathbb{E}[\|\mathbf{b}_t^{(k)} - \bar{\mathbf{b}}_t\|^2] \,.
\end{aligned}
$$
$$\tag{47}$$

According to the value of $\{C_i\}_{i=0}^8$ in Eq. (14) and Lemma 11, it is easy to get

$$
\begin{aligned}
H_{t+1} - H_t \leq &-\frac{\gamma_1\eta}{2}\mathbb{E}[\|\nabla\Phi(\bar{\mathbf{x}}_t)\|^2] - \frac{\gamma_1\eta L^2}{2}\mathbb{E}[\|\bar{\mathbf{y}}_t - \mathbf{y}^*(\bar{\mathbf{x}}_t)\|^2] \\
&+ \Big[3\gamma_1^2\eta^2\Big(\frac{13\gamma_1\eta L^2}{s_1\rho_1} + \frac{1006\eta\gamma_1 L^2}{s_1\rho_1} + (1-\frac{s_1}{n})(1+\frac{1}{\alpha_1})\frac{28n\rho_1\gamma_1\eta L^2}{s_1^2} + (1-\frac{s_1}{n})(1+\frac{1}{\alpha_1})\frac{2024n\rho_1\eta\gamma_1 L^2}{s_1^2}\Big) \\
&+ \frac{25\eta\gamma_1^2\kappa^2}{6\gamma_2\mu}C_0 - \frac{\gamma_1\eta}{4}\Big]\mathbb{E}[\|\bar{\mathbf{v}}_t\|^2] \\
&+ \Big[3\gamma_2^2\eta^2\Big(\frac{13\gamma_1\eta L^2}{s_1\rho_1} + \frac{1006\eta\gamma_1 L^2}{s_1\rho_1} + (1-\frac{s_1}{n})(1+\frac{1}{\alpha_1})\frac{28n\rho_1\gamma_1\eta L^2}{s_1^2} + (1-\frac{s_1}{n})(1+\frac{1}{\alpha_1})\frac{2024n\rho_1\eta\gamma_1 L^2}{s_1^2}\Big) \\
&- \frac{3\gamma_2^2\eta}{4}C_0\Big]\mathbb{E}[\|\bar{\mathbf{u}}_t\|^2] \\
&+ \Big[12\eta^2\Big(\frac{13\gamma_1\eta L^2}{s_1\rho_1} + \frac{1006\eta\gamma_1 L^2}{s_1\rho_1} + (1-\frac{s_1}{n})(1+\frac{1}{\alpha_1})\frac{28n\rho_1\gamma_1\eta L^2}{s_1^2} + (1-\frac{s_1}{n})(1+\frac{1}{\alpha_1})\frac{2024n\rho_1\eta\gamma_1 L^2}{s_1^2}\Big) \\
&+ 2\gamma_1\eta L^2 + \frac{25\eta\gamma_2 L^2}{3\mu}C_0 - \frac{\eta(1-\lambda^2)}{2}C_5\Big]\frac{1}{K}\sum_{k=1}^K\mathbb{E}[\|\bar{\mathbf{x}}_t - \mathbf{x}_t^{(k)}\|^2] \\
&+ \Big[12\eta^2\Big(\frac{13\gamma_1\eta L^2}{s_1\rho_1} + \frac{1006\eta\gamma_1 L^2}{s_1\rho_1} + (1-\frac{s_1}{n})(1+\frac{1}{\alpha_1})\frac{28n\rho_1\gamma_1\eta L^2}{s_1^2} + (1-\frac{s_1}{n})(1+\frac{1}{\alpha_1})\frac{2024n\rho_1\eta\gamma_1 L^2}{s_1^2}\Big) \\
&+ 2\gamma_1\eta L^2 + \frac{25\eta\gamma_2 L^2}{3\mu}C_0 - \frac{\eta(1-\lambda^2)}{2}C_6\Big]\frac{1}{K}\sum_{k=1}^K\mathbb{E}[\|\bar{\mathbf{y}}_t - \mathbf{y}_t^{(k)}\|^2] \\
&+ \Big[3\gamma_1^2\eta^2\Big(\frac{13\gamma_1\eta L^2}{s_1\rho_1} + \frac{1006\eta\gamma_1 L^2}{s_1\rho_1} + (1-\frac{s_1}{n})(1+\frac{1}{\alpha_1})\frac{28n\rho_1\gamma_1\eta L^2}{s_1^2} + (1-\frac{s_1}{n})(1+\frac{1}{\alpha_1})\frac{2024n\rho_1\eta\gamma_1 L^2}{s_1^2}\Big) \\
&+ \frac{2\gamma_1^2\eta}{1-\lambda^2}C_5 - \frac{1-\lambda^2}{2}C_7\Big]\frac{1}{K}\sum_{k=1}^K\|\mathbf{a}_t^{(k)} - \bar{\mathbf{a}}_t\|^2] \\
&+ \Big[3\gamma_2^2\eta^2\Big(\frac{13\gamma_1\eta L^2}{s_1\rho_1} + \frac{1006\eta\gamma_1 L^2}{s_1\rho_1} + (1-\frac{s_1}{n})(1+\frac{1}{\alpha_1})\frac{28n\rho_1\gamma_1\eta L^2}{s_1^2} + (1-\frac{s_1}{n})(1+\frac{1}{\alpha_1})\frac{2024n\rho_1\eta\gamma_1 L^2}{s_1^2}\Big) \\
&+ \frac{2\eta\gamma_2^2}{1-\lambda^2}C_6 - \frac{1-\lambda^2}{2}C_8\Big]\frac{1}{K}\sum_{k=1}^K\mathbb{E}[\|\mathbf{b}_t^{(k)} - \bar{\mathbf{b}}_t\|^2]
\end{aligned}
\tag{48}
$$

According to the value of $\{C_i\}_{i=0}^8$ in Eq. (14) and the value of $\gamma_1$, $\gamma_2$, $\eta$ in Theorem 1, it is easy to get

$$
H_{t+1} - H_t \leq -\frac{\gamma_1\eta}{2}\mathbb{E}[\|\nabla\Phi(\bar{\mathbf{x}}_t)\|^2] - \frac{\gamma_1\eta L^2}{2}\mathbb{E}[\|\bar{\mathbf{y}}_t - \mathbf{y}^*(\bar{\mathbf{x}}_t)\|^2] \ .
\tag{49}
$$

Then, by summing $t$ from 1 to $T-1$, we can get

$$\sum_{t=1}^{T-1} \frac{\gamma_1 \eta}{2} \mathbb{E}[\|\nabla\Phi(\bar{\mathbf{x}}_t)\|^2] + \frac{\gamma_1 \eta L^2}{2} \mathbb{E}[\|\bar{\mathbf{y}}_t - \mathbf{y}^*(\bar{\mathbf{x}}_t)\|^2] \leq H_1 - \Phi(\mathbf{x}_*)$$

$$\leq \Phi(\bar{\mathbf{x}}_0) - \Phi(\mathbf{x}_*) - \frac{\gamma_1 \eta}{2}\|\nabla\Phi(\bar{\mathbf{x}}_0)\|^2 + \left(\gamma_1 \eta L^2 + (1 - \frac{\eta\gamma_2\mu}{4})\frac{6\gamma_1 L^2}{\gamma_2\mu}\right)\mathbb{E}[\|\bar{\mathbf{y}}_0 - \mathbf{y}^*(\bar{\mathbf{x}}_0)\|^2]$$

$$+ \left(-\frac{3\gamma_2^2 \eta}{4}\frac{6\gamma_1 L^2}{\gamma_2\mu}\right)\mathbb{E}[\|\bar{\mathbf{u}}_0\|^2] + \left(\frac{25\eta\gamma_1^2\kappa^2}{6\gamma_2\mu}\frac{6\gamma_1 L^2}{\gamma_2\mu} - \frac{\gamma_1\eta}{4}\right)\mathbb{E}[\|\bar{\mathbf{v}}_0\|^2]$$

$$+ \left(2\gamma_1\eta + \frac{3\gamma_1\eta}{\rho_1}(1-\rho_1)^2 + \frac{6\rho_1^2}{1-\lambda^2}\frac{(1-\lambda^2)\gamma_1\eta}{6\rho_1}\right)\frac{1}{K}\sum_{k=1}^K \mathbb{E}[\|\nabla_{\mathbf{x}} f(\mathbf{x}_0^{(k)}, \mathbf{y}_0^{(k)}) - \mathbf{v}_0^{(k)}\|^2]$$

$$+ \left(\frac{25\eta\gamma_2}{3\mu}\frac{6\gamma_1 L^2}{\gamma_2\mu} + \frac{250\eta\gamma_1 L^2}{\rho_1\mu^2}(1-\rho_1)^2 + \frac{6\rho_1^2}{1-\lambda^2}\frac{(1-\lambda^2)\eta\gamma_1 L^2}{\rho_1\mu^2}\right)\frac{1}{K}\sum_{k=1}^K \mathbb{E}[\|\nabla_{\mathbf{y}} f^{(k)}(\mathbf{x}_0^{(k)}, \mathbf{y}_0^{(k)}) - \mathbf{u}_0^{(k)}\|^2]$$

$$+ \left(\frac{14n\rho_1\gamma_1\eta}{s_1^2}(1-\frac{s_1}{n})(1+\alpha_1) + \frac{2\rho_1^2}{s_1}\frac{3\gamma_1\eta}{\rho_1} + \frac{6\rho_1^2}{(1-\lambda^2)s_1}\frac{(1-\lambda^2)\gamma_1\eta}{6\rho_1}\right)\frac{1}{K}\sum_{k=1}^K \frac{1}{n}\sum_{i=1}^n \mathbb{E}[\|\nabla_{\mathbf{x}} f_i(\mathbf{x}_0^{(k)}, \mathbf{y}_0^{(k)}) - \mathbf{g}_{i,0}^{(k)}\|^2]$$

$$+ \left(\frac{1012n\rho_1\eta\gamma_1 L^2}{s_1^2\mu^2}(1-\frac{s_1}{n})(1+\alpha_1) + \frac{2\rho_1^2}{s_1}\frac{250\eta\gamma_1 L^2}{\rho_1\mu^2} + \frac{6\rho_1^2}{(1-\lambda^2)s_1}\frac{(1-\lambda^2)\eta\gamma_1 L^2}{\rho_1\mu^2}\right)\frac{1}{K}\sum_{k=1}^K \frac{1}{n}\sum_{i=1}^n \mathbb{E}[\|\nabla_{\mathbf{y}} f_i(\mathbf{x}_0^{(k)}, \mathbf{y}_0^{(k)}) - \mathbf{h}_{i,0}^{(k)}\|^2]$$

$$+ \left(\frac{4L^2}{s_1}\frac{250\eta\gamma_1 L^2}{\rho_1\mu^2} + \frac{4L^2}{s_1}\frac{3\gamma_1\eta}{\rho_1} + 2L^2(1-\frac{s_1}{n})(1+\frac{1}{\alpha_1})\frac{14n\rho_1\gamma_1\eta}{s_1^2} + 2L^2(1-\frac{s_1}{n})(1+\frac{1}{\alpha_1})\frac{1012n\rho_1\eta\gamma_1 L^2}{s_1^2\mu^2}\right.$$

$$\left. + \frac{6L^2}{(1-\lambda^2)s_1}\frac{(1-\lambda^2)\gamma_1\eta}{6\rho_1} + \frac{6L^2}{(1-\lambda^2)s_1}\frac{(1-\lambda^2)\eta\gamma_1 L^2}{\rho_1\mu^2}\right)\frac{1}{K}\sum_{k=1}^K \mathbb{E}[\|\mathbf{x}_1^{(k)} - \mathbf{x}_0^{(k)}\|^2]$$

$$+ \left(\frac{4L^2}{s_1}\frac{250\eta\gamma_1 L^2}{\rho_1\mu^2} + \frac{4L^2}{s_1}\frac{3\gamma_1\eta}{\rho_1} + 2L^2(1-\frac{s_1}{n})(1+\frac{1}{\alpha_1})\frac{14n\rho_1\gamma_1\eta}{s_1^2} + 2L^2(1-\frac{s_1}{n})(1+\frac{1}{\alpha_1})\frac{1012n\rho_1\eta\gamma_1 L^2}{s_1^2\mu^2}\right.$$

$$\left. + \frac{6L^2}{(1-\lambda^2)s_1}\frac{(1-\lambda^2)\gamma_1\eta}{6\rho_1} + \frac{6L^2}{(1-\lambda^2)s_1}\frac{(1-\lambda^2)\eta\gamma_1 L^2}{\rho_1\mu^2}\right)\frac{1}{K}\sum_{k=1}^K \mathbb{E}[\|\mathbf{y}_1^{(k)} - \mathbf{y}_0^{(k)}\|^2]$$

$$+ \left(\frac{2\gamma_1^2\eta}{1-\lambda^2}\frac{106\gamma_1\kappa^2 L^2}{(1-\lambda^2)} + \frac{1+\lambda^2}{2}\frac{(1-\lambda^2)\gamma_1\eta}{6\rho_1}\right)\frac{1}{K}\sum_{k=1}^K \mathbb{E}[\|\mathbf{a}_0^{(k)} - \bar{\mathbf{a}}_0\|^2]$$

$$+ \left(\frac{2\eta\gamma_2^2}{1-\lambda^2}\frac{106\gamma_1\kappa^2 L^2}{(1-\lambda^2)} + \frac{1+\lambda^2}{2}\frac{(1-\lambda^2)\eta\gamma_1 L^2}{\rho_1\mu^2}\right)\frac{1}{K}\sum_{k=1}^K \mathbb{E}[\|\mathbf{b}_0^{(k)} - \bar{\mathbf{b}}_0\|^2] .$$

$$(50)$$

Additionally, due to

$$\frac{1}{K}\sum_{k=1}^K \mathbb{E}[\|\mathbf{a}_0^{(k)} - \bar{\mathbf{a}}_0\|^2] = \frac{1}{K}\sum_{k=1}^K \mathbb{E}[\|\mathbf{v}_0^{(k)} - \frac{1}{K}\sum_{k=1}^K \mathbf{v}_0^{(k)}\|^2]$$

$$= \frac{1}{K}\sum_{k=1}^K \mathbb{E}[\|\mathbf{v}_0^{(k)} - \nabla_{\mathbf{x}} f(\mathbf{x}_0^{(k)}, \mathbf{y}_0^{(k)}) + \nabla_{\mathbf{x}} f(\mathbf{x}_0^{(k)}, \mathbf{y}_0^{(k)}) - \frac{1}{K}\sum_{k=1}^K \nabla_{\mathbf{x}} f(\mathbf{x}_0^{(k)}, \mathbf{y}_0^{(k)}) + \frac{1}{K}\sum_{k=1}^K \nabla_{\mathbf{x}} f(\mathbf{x}_0^{(k)}, \mathbf{y}_0^{(k)}) - \frac{1}{K}\sum_{k=1}^K \mathbf{v}_0^{(k)}\|^2]$$

$$= \frac{1}{K}\sum_{k=1}^K \mathbb{E}[\|\mathbf{v}_0^{(k)} - \nabla_{\mathbf{x}} f(\mathbf{x}_0^{(k)}, \mathbf{y}_0^{(k)})\|^2] + \frac{1}{K}\sum_{k=1}^K \mathbb{E}[\|\nabla_{\mathbf{x}} f(\mathbf{x}_0^{(k)}, \mathbf{y}_0^{(k)}) - \frac{1}{K}\sum_{k=1}^K \nabla_{\mathbf{x}} f(\mathbf{x}_0^{(k)}, \mathbf{y}_0^{(k)})\|^2]$$

$$+ \frac{1}{K}\sum_{k=1}^K \mathbb{E}[\|\frac{1}{K}\sum_{k=1}^K \nabla_{\mathbf{x}} f(\mathbf{x}_0^{(k)}, \mathbf{y}_0^{(k)}) - \frac{1}{K}\sum_{k=1}^K \mathbf{v}_0^{(k)}\|^2]$$

$$\leq \frac{2}{K}\sum_{k=1}^K \mathbb{E}[\|\mathbf{v}_0^{(k)} - \nabla_{\mathbf{x}} f(\mathbf{x}_0^{(k)}, \mathbf{y}_0^{(k)})\|^2] ,$$

$$(51)$$

we can get

$$\frac{1}{K}\sum_{k=1}^{K}\mathbb{E}[\|\mathbf{x}_1^{(k)} - \mathbf{x}_0^{(k)}\|^2] = \frac{1}{K}\mathbb{E}[\|X_1 - X_0\|_F^2]$$

$$\leq 12\eta^2\frac{1}{K}\sum_{k=1}^{K}\mathbb{E}[\|\mathbf{x}_0^{(k)} - \bar{\mathbf{x}}_0\|^2] + 3\gamma_1^2\eta^2\frac{1}{K}\sum_{k=1}^{K}\mathbb{E}[\|\mathbf{a}_0^{(k)} - \bar{\mathbf{a}}_0\|^2] + 3\gamma_1^2\eta^2\mathbb{E}[\|\bar{\mathbf{v}}_0\|^2]$$

$$= 3\gamma_1^2\eta^2\frac{1}{K}\sum_{k=1}^{K}\mathbb{E}[\|\mathbf{a}_0^{(k)} - \bar{\mathbf{a}}_0\|^2] + 3\gamma_1^2\eta^2\mathbb{E}[\|\bar{\mathbf{v}}_0\|^2] \tag{52}$$

$$= 6\gamma_1^2\eta^2\frac{1}{K}\sum_{k=1}^{K}\mathbb{E}[\|\mathbf{v}_0^{(k)} - \nabla_{\mathbf{x}}f(\mathbf{x}_0^{(k)}, \mathbf{y}_0^{(k)})\|^2]] + 3\gamma_1^2\eta^2\mathbb{E}[\|\bar{\mathbf{v}}_0\|^2] .$$

Similarly, we can get

$$\frac{1}{K}\sum_{k=1}^{K}\mathbb{E}[\|\mathbf{y}_1^{(k)} - \mathbf{y}_0^{(k)}\|^2] = 6\gamma_2^2\eta^2\frac{1}{K}\sum_{k=1}^{K}\mathbb{E}[\|\mathbf{u}_0^{(k)} - \nabla_{\mathbf{y}}f(\mathbf{x}_0^{(k)}, \mathbf{y}_0^{(k)})\|^2] + 3\gamma_2^2\eta^2\mathbb{E}[\|\bar{\mathbf{u}}_0\|^2] . \tag{53}$$

By plugging the last two inequalities into Eq. 50, we can get

$$\sum_{t=1}^{T-1}\frac{\gamma_1\eta}{2}\mathbb{E}[\|\nabla\Phi(\bar{\mathbf{x}}_t)\|^2] + \frac{\gamma_1\eta L^2}{2}\mathbb{E}[\|\bar{\mathbf{y}}_t - \mathbf{y}^*(\bar{\mathbf{x}}_t)\|^2]$$

$$\leq \Phi(\bar{\mathbf{x}}_0) - \Phi(\mathbf{x}_*) - \frac{\gamma_1\eta}{2}\|\nabla\Phi(\bar{\mathbf{x}}_0)\|^2 + \left(\frac{6\gamma_1 L^2}{\gamma_2\mu} - \frac{1}{2}\gamma_1\eta L^2\right)\|\bar{\mathbf{y}}_0 - \mathbf{y}^*(\bar{\mathbf{x}}_0)\|^2$$

$$+ \gamma_1\eta\kappa^2\frac{2(n-s_0)}{s_0 s_1}\left(7 + 15369L^2\gamma_1^2\eta^2 + \frac{424\gamma_1^2 L^2}{(1-\lambda^2)^2}\right)\frac{1}{K}\sum_{k=1}^{K}\frac{1}{n}\sum_{i=1}^{n}\|\nabla_{\mathbf{x}}f_i(\mathbf{x}_0^{(k)}, \mathbf{y}_0^{(k)})\|^2$$

$$+ \eta\gamma_1\kappa^2\frac{2(n-s_0)}{s_0 s_1}\left(258 + 50L^2 + 15369L^2\gamma_2^2\eta^2 + \frac{424\gamma_2^2 L^2}{(1-\lambda^2)^2}\right)\frac{1}{K}\sum_{k=1}^{K}\frac{1}{n}\sum_{i=1}^{n}\|\nabla_{\mathbf{y}}f_i(\mathbf{x}_0^{(k)}, \mathbf{y}_0^{(k)})\|^2 \tag{54}$$

$$+ 14\gamma_1\eta\frac{s_0 - s_0^2/n}{2s_0 s_1}\frac{1}{K}\sum_{k=1}^{K}\frac{1}{n}\sum_{j=1}^{n}\|\nabla_{\mathbf{x}}f_j(\mathbf{x}_0^{(k)}, \mathbf{y}_0^{(k)})\|^2$$

$$+ 1012\kappa^2\eta\gamma_1\frac{s_0 - s_0^2/n}{2s_0 s_1}\frac{1}{K}\sum_{k=1}^{K}\frac{1}{n}\sum_{j=1}^{n}\|\nabla_{\mathbf{y}}f_j(\mathbf{x}_0^{(k)}, \mathbf{y}_0^{(k)})\|^2 ,$$

where the last step follows from $\rho_1 = \frac{s_1}{2n}$. By dividing $\frac{\gamma_1\eta}{2}$ on both sides, we can get

$$\frac{1}{T}\sum_{t=0}^{T-1}(\mathbb{E}[\|\nabla\Phi(\bar{\mathbf{x}}_t)\|^2] + L^2\mathbb{E}[\|\bar{\mathbf{y}}_t - \mathbf{y}^*(\bar{\mathbf{x}}_t)\|^2])$$

$$\leq \frac{2(\Phi(\bar{\mathbf{x}}_0) - \Phi(\mathbf{x}_*))}{\gamma_1\eta T} + \frac{12\kappa L}{\gamma_2\eta T}\|\bar{\mathbf{y}}_0 - \mathbf{y}^*(\bar{\mathbf{x}}_0)\|^2$$

$$+ \left(\frac{4\kappa^2(n-s_0)}{s_0 s_1}\left(7 + 15369L^2\gamma_1^2 + \frac{424\gamma_1^2 L^2}{(1-\lambda^2)^2}\right) + \frac{28(s_0 - s_0^2/n)}{2s_0 s_1}\right)\frac{1}{TK}\sum_{k=1}^{K}\frac{1}{n}\sum_{i=1}^{n}\|\nabla_{\mathbf{x}}f_i(\mathbf{x}_0^{(k)}, \mathbf{y}_0^{(k)})\|^2$$

$$+ \left(\frac{4\kappa^2(n-s_0)}{s_0 s_1}\left(258 + 50L^2 + 15369L^2\gamma_2^2 + \frac{424\gamma_2^2 L^2}{(1-\lambda^2)^2}\right) + \frac{2024\kappa^2(s_0 - s_0^2/n)}{2s_0 s_1}\right)\frac{1}{TK}\sum_{k=1}^{K}\frac{1}{n}\sum_{i=1}^{n}\|\nabla_{\mathbf{y}}f_i(\mathbf{x}_0^{(k)}, \mathbf{y}_0^{(k)})\|^2 , \tag{55}$$

which completes the proof.

$\square$

