# OpenReview forum: "Decentralized Stochastic Gradient Descent Ascent  for Finite-Sum Minimax Problems"
_TMLR — Rejected by TMLR_

### Review · Reviewer_RmHB · 2023-12-14

**Summary Of Contributions:**

This paper studies the problem of finite-sum minimax optimization, which has received much attention recently. More specifically, the authors focus on the distributed setting, where the training data is distributed among multiple workers, and develop a decentralized algorithm, which improves the sample complexity of the current best one by removing an additive factor of $n$, where $n$ is the number of data on each worker. Experimental results are provided to verify the effectiveness and efficiency of the proposed method.

**Audience:**

Yes

**Broader Impact Concerns:**

None.

**Claims And Evidence:**

Yes

**Requested Changes:**

1) Please highlight the novelty of the proposed algorithm.
2) Please discuss the space cost of the proposed algorithm.
3) It would be better if the authors conduct some experiments on imbalanced data.

**Strengths And Weaknesses:**

Strengths:
1) The studied problem is interesting, and this paper is well written.
2) The proposed algorithm can improve the sample complexity of the current best one by removing an additive factor of $n$, where $n$ is the number of data on each worker.
3) The theoretical analysis is easy to follow, and the experimental results are sound.

Weaknesses:
1) The novelty of the proposed algorithm is limited. The key technique in the proposed algorithm is a variance-reduced gradient estimator without querying the full gradient which was proposed by Li & Richtárik (2021). Although Li & Richtárik (2021) only considered the minimization problem in the centralized setting, it seems not hard to extend the result to the decentralized setting.
2) The space complexity of the proposed algorithm is very high, i.e., $O(n)$, instead of $O(1)$ of existing algorithms. Notice that when $O(n)$ space is allowed, the idea of variance reduction without querying the full gradient can date back to Roux et al. (2012), which was missed by the authors.
3) In Subsection 5.2, the authors mentioned that AUC maximization is a commonly used method for the imbalanced data classification problem. However, the data used in the experiment is not imbalanced, which makes the experiment less persuasive.

[1] N. L. Roux, M. Schmidt, and F. Bach. A stochastic gradient method with an exponential convergence rate for finite training sets. In NIPS, pages 2672–2680, 2012.

---

### Review · Reviewer_fgy4 · 2024-01-11

**Summary Of Contributions:**

This paper proposes a novel variance-reduced algorithm for minimax problem in a federated learning scenario. The algorithm (cf. page 4) appears to be a combination of various techniques from the literature, including the variance reduced gradient as in [1], together with some averaging tricks such as the ones in lines 4 and 5 in the algorithm. There is also a separate quantity $g$ (and its corresponding $h$) which keep track of all the gradients witnessed by any of the workers.

The main contribution is then a proof that the above algorithm achieves superior rates to the state of the art in terms of sample complexity and communication complexity. Furthermore, experiments on AUC maximization via a minimax surrogate (as per [2]) show that the proposed algorithm significantly outperforms other optimization strategies.










===============================

References:


[1] Zhize Li and Peter Richtárik. Zerosarah: Efficient nonconvex finite-sum optimization with zero full gradient computation.


[2] Yiming Ying, Longyin Wen, Siwei Lyu. Stochastic Online AUC Maximization.  NeurIPS 2016.


[3] Feihu Huang, Shangqian Gao, and Heng Huang. Gradient descent ascent for min-max problems on Riemannian manifolds

**Audience:**

Yes

**Claims And Evidence:**

Yes

**Requested Changes:**

**Major**:

1. Please show more details for the proof of equation 15/46. This should definitely be a lemma on its own.

2. I have a perhaps naive question regarding the communication complexity: first of all, I would like the authors to define it properly and show how it can be computed from Corollary 1. Secondly, it seems that the left hand side of Corollary 1 involves an average over all the workers (inside the definition of quantities such as $\bar{x}_t$). Why does that not imply an additional communication round where all the workers communicate with each other? I understand that perhaps one can argue that Lemma 9 independently implies that the LHS of corollary 1 is close enough to any specific worker, but it seems that this calculation should be incorporated into a more specific statement inside corollary 1 (and it is not clear if it would change the sample complexity or the requirements on the learning parameters).

3. In theorem 1, it is stated that $\phi$ is $L_{\phi}=2\kappa L$ smooth.

3.1 (more minor) This is stated without proof or even a reference. I would like to see a proof. It is counterintuitive to me that $\mu$ is not involved in the formula.

3.2 (major) As far as I understand, the condition on $\eta$ in Lemma 1 should be $\eta\leq \frac{1}{2\gamma_1 L_{\phi}}=\frac{1}{2\kappa \gamma_1 \kappa L}$, but instead, it is stated in terms of $L$ (not $L_\phi$). This appears to be a relatively **serious problem** which could cause errors to spread throughout the whole proofs, potentially modifying the values of all of the other parameters by relevant factors of $\kappa$. Indeed, the L in the first term of the definition of $H_t$ should be $L_{\phi}$, but not the other Ls appearing (e.g.) in the third and fourth terms.
I understand that the final power of $\kappa$  (i.e., 3) in the main result is the same as in related works, so it is very likely that all of this can be fixed without issue, but the details should be written down more properly.


More minor issues or  **questions**:

1. I found the statement of Theorem 1 somewhat confusing in terms of the definitions of $\bar{x}$ and $y^*$. It appears that all of these quantities systematically involve dependence on $t$, so that when the authros write $\phi^{(k)}(x)=\sum ... f(x,y^*)$, the $y^*$ actually refers to the quantity $y^*(\bar{x}_t)$ where $\bar{x}_t$ is itself defined earlier in the line. This seems to be confirmed in calculations such as that of equation (19). However, to a reader unfamiliar with the related works, it appears that $y^*$ could refer to some sort of global minimum.

2. A similar issue occurs on page 21 (at the top), where the $\phi(x_*)$ is a little mysterious: it is not clear  whether it in fact refers to $x^*_t$, how it is defined, or how it can be shown to appear there based on the previous line.


3. Tried to prove equation 15 a bit, by checking some of the terms, and although some of them work out fine, some of them appear more cryptic. For instance, in the quantity which multiplies$\frac{1}{K}\sum_{k=1}^K \|\nabla_y f_i(x_t^{k}, x_t^{k})-u_t^{(k)}\|^2$ , I can see how one gets a  $\frac{250 \eta \gamma_1}L^2{\rho_1 \mu^2} [(1-\rho_1)^2-1]$ from the definition of the potential function and an application of Lemma 6, and then also a factor of $ \frac{6\eta L^2}{\gamma_2 \mu} \frac{25\eta \gamma_2}{\mu}$, but I can't see where the $\frac{6\rho_1^2}{1-\lambda^2} \frac{(1-\lambda^2) \gamma_1 \eta }{\rho_1 \mu^2}$ comes from. Can you help me?
It also seems like some of the calculations on 21 should actually be included in a proof of equation 46, but this is not explained clearly.

4. In the bottom of page 16, I really can't manage to get the correct factor of $\frac{n-s_0}{(n-1)s_0}$. Could you add more details?

5. (important) In the proof o lemma 3, I am having a lot of trouble getting to the $\frac{2}{1-\lambda^2}$ at the next to last line. It seems that $1+1/a =\frac{1+\lambda^2}{1-\lambda^2}$ instead.

6 In page 14, it is mentioned that the first equation in the proof of Lemma 2 is "from [3]", but there is no mention of which part of the paper this equation is taken from.






Very minor (typos) etc.

1. At the top of the appendix, the explanation of the generic construction of vectors containing any quantity evaluated for each worker has an unclosed square bracket. This definition should also be highlighted a bit more, since the beginning of the proof of Lemma 3 is very confusing unless one has really absorbed that (presumably standard) convention.

2. There are two equal signs instead of 1 in the indicator function at the bottom of page 17

3. The use of the word "totally" after page 1 is not appropriate. I think the authors mean "in total".

4. At the bottom of page 20, "the value of $\gamma_1,\gamma_2, n$" should be ""the values of $\gamma_1,\gamma_2, n$"

**Strengths And Weaknesses:**

It is clear that this is a **very strong paper** which I think should be accepted after being made a bit more reader friendly and correcting minor errors.

Strengths:

1.There appears to be a non trivial improvement in the rate provided for this important problem.

2. The proofs are highly computationally non trivial.

3. The execution is generally very clean and professional.

4. The experimental results are very compelling.

5. The motivating application of AUC maximization and the surrogate problem provided is very engaging.


Weaknesses:

1. A couple of typos or minor errors.

2. The mathematical proofs are not reader friendly at all. I am not that familiar with the associated classic techniques so it can be hard for me to follow. In many cases, this can safely be reduced to my own issue (lack of familiarity with the related works), but in some cases, I think it is safe to say that the paper is too laconic. For instance, the "proof" of equation 15/46 on page 20  just says "according to the afore mentioned lemmas, it is easy to get the following". It is not easy at all, it could easily take a good reader hours to check everything there. Note that this is one of the only places where the motivation for the complicated expressions for $\gamma_1,\gamma_2$ and similar learning parameters from Theorem 1 appear. Given the lack of details, it seems statistically extremely unlikely that the associated expressions in Theorem are all correct.


3. It is hard for me to judge novelty due to the lack of familiarity with the related works. For instance, I can see that [1] is acknowledged for the variance reduced version of the gradient, but I don't know to what extent the other components of the model are standard or known. For instance, is the $g_t$ a standard quantity? Likewise, it seems like the proof in equation (24) is probably very similar to calculations in [1].

---

### Review · Reviewer_6hsa · 2024-01-16

**Summary Of Contributions:**

A novel algorithm is introduced to address the distributed stochastic nonconvex-strongly concave minimax problem. Leveraging variance-reduced techniques, this new method demonstrates an improved sample complexity.

**Audience:**

Yes

**Claims And Evidence:**

Yes

**Requested Changes:**

1. As you have discussed in the introduction, there is a lot of work studying the nonconvex strongly-concave minimax problems. To offer a more comprehensive understanding, it would be advantageous to provide additional context on the theoretical challenges that arise  specifically in the decentralized setting.

2. I suggest adding three baselines: GT-SRVRI [3], DSVRGDA-P, and DSVRGDA-Z [2]. Given the superior performance of GT-SRVRI compared to GT-SRVR, including it for comparison would provide a more comprehensive assessment. Additionally, since DSVRGDA-Z also employs the variance-reduced gradient estimator technique, a comparison with DSVRGDA-Z would enhance the evaluation's depth and relevance.

[3] Zhang, X., Liu, Z., Liu, J., Zhu, Z., & Lu, S. (2021). Taming communication and sample complexities in decentralized policy evaluation for cooperative multi-agent reinforcement learning. Advances in Neural Information Processing Systems, 34, 18825-18838.

**Strengths And Weaknesses:**

**Strengths**:

1. The sample complexity of the new algorithm is better than existing methods.
2. Experiments on the AUC maximization problem have been done to demonstrate the efficacy of the new proposed algorithm.

**Weakness**:
1. The newly proposed algorithm DSGDA applies the idea from [1] of computing a gradient estimator with reduced variance to a decentralized minimax problem. Similar ideas have also been used in [2]. Can you elaborate on the main advantages of your proposed new algorithm compared to [2]?
2. I am more interested in understanding which step primarily contributes to enhancing sample complexity: the variance-reduced gradient estimator (Equation 2) or the momentum step (Equation 7).

[1] Li, Z., Hanzely, S., & Richtárik, P. (2021). ZeroSARAH: Efficient nonconvex finite-sum optimization with zero full gradient computation. arXiv preprint arXiv:2103.01447.

[2] Zhang, Y., Jiang, W., Zheng, F., Tan, C. C., Shi, X., & Gao, H. (2023). Can Decentralized Stochastic Minimax Optimization Algorithms Converge Linearly for Finite-Sum Nonconvex-Nonconcave Problems?. arXiv preprint arXiv:2304.11788.

---

### Decision · Action_Editor_UkXf · 2024-02-25

**Recommendation:** Reject

**Comment:**

The paper receives three reviews. The reviewers acknowledge that the proposed algorithm has improved sample complexity and has achieved convincing performance on experiments. However, several concerns have been proposed. One reviewer indicates the comparison with the NC-PL setting is needed. Another reviewer has concerns on the space complexity.

I went through the paper and found some issues on the theoretical analysis. For example, in the last step of Eq. (24), the authors use the inequality

$$
E\Big\|\frac{1}{s}\sum_i\nabla f_i(x_t^k,y_t^k)-g_{i,t}^k+\frac{1}{n}\sum_jg_{j,t}^k-\nabla f_j(x_t^k,y_t^k)\Big\|^2
\leq \frac{1}{ns}\sum_{j=1}^{n} E\Big\|\nabla f_j(x_t^k,y_t^k)-g_{j,t}^k\Big\|^2
$$

It seems that this inequality is problematic. There should not be a factor of $1/(ns)$ in the right hand side. The reason is that the right-hand side decays of the order of $1/s$, while the left-hand side should be of order of $1$.

If Eq (24) is problematic, then Lemma 3 and Lemma 4 may not hold. Then the sample complexity analysis is not very rigorous since Lemma 3 and Lemma 4 are used in the sample analysis.

There are several notational typos in the analysis:
- Eq (2), $f_i$ should be $f_i^{(k)}$. This also happens at several places in the proof, e.g., proof of Lemma 1, Lemma 3.
- In Eq (4): the authors use $f(x_t^k,y_t^k)$. However, $f$ is not defined. Should this be $f_j(x_t^k,y_t^k)$
- Eq (23): $\bar{A}_t$ should be $\bar{A}_tW$ in several places.

Based on the comments from the reviewers and the above concern on the theoretical analysis, I would not recommend publishing the paper in its current form. The authors are welcome to submit a revision addressing these comments in the future.

**Audience:**

Yes. The paper considers an important class of machine learning problems with minimax structure, which have many applications in practice.

**Claims And Evidence:**

The paper studies decentralized minimax optimization problems of finite-sum structure. The paper proposes an efficient decentralized stochastic gradient descent ascent algorithm, where the workers are connected through a communication network. Each work updates the model based on its own dataset and communicates with neighboring workers. The paper studies the sample complexity and communication complexity of the algorithm for nonconvex-strongly-concave problems.

The proposed algorithm has improved sample complexity as compared to existing decentralized minimax problems. However, the theoretical analysis seems to have flaws. Details are given below.

Experimental comparisons with several existing methods on AUC maximization problems are also given.

**Resubmission Of Major Revision:**

The authors may consider submitting a major revision at a later time.

---

> ### Author Response · Authors · 2024-02-27
> **We respectfully disagree with the error pointed out by Action Editor UkXf**
>
> We appreciate the action editor's time and constructive suggestions. We will revise those typos. However, we respectfully disagree with the error pointed out by AE.  Below is the proof of that inequality.
>
>
>
> $\mathbb{E}[\|\frac{1}{s} \sum\_{i \in \mathcal{S}}\Big(\nabla\_{\mathbf{x}} f^{(k)}\_{i}(\mathbf{x}\_{t}^{(k)}, \mathbf{y}\_{t}^{(k)})-\mathbf{g}\_{i, t}^{(k)}\Big)+\frac{1}{n} \sum_{j=1}^{n} \mathbf{g}\_{j, t}^{(k)}- \nabla_{\mathbf{x}} f_j^{(k)}(\mathbf{x}\_{t}^{(k)}, \mathbf{y}\_{t}^{(k)}) \|^2]$
>
> $ = \mathbb{E}[\|\frac{1}{s} \sum\_{i \in \mathcal{S}}\Big(\nabla\_{\mathbf{x}} f^{(k)}\_{i}(\mathbf{x}\_{t}^{(k)}, \mathbf{y}\_{t}^{(k)})-\mathbf{g}\_{i, t}^{(k)}+\frac{1}{n} \sum_{j=1}^{n} \mathbf{g}\_{j, t}^{(k)}- \frac{1}{n} \sum_{j=1}^{n} \nabla_{\mathbf{x}} f_j^{(k)}(\mathbf{x}\_{t}^{(k)}, \mathbf{y}\_{t}^{(k)}) \Big)\|^2] $
>
> $ =\frac{1}{s^2} \sum\_{i \in \mathcal{S}}\mathbb{E}[\|\nabla\_{\mathbf{x}} f^{(k)}\_{i}(\mathbf{x}\_{t}^{(k)}, \mathbf{y}\_{t}^{(k)})-\mathbf{g}\_{i, t}^{(k)}+\frac{1}{n} \sum_{j=1}^{n} \mathbf{g}\_{j, t}^{(k)}- \frac{1}{n} \sum_{j=1}^{n} \nabla_{\mathbf{x}} f_j^{(k)}(\mathbf{x}\_{t}^{(k)}, \mathbf{y}\_{t}^{(k)}) \|^2] $
>
> $\leq \frac{1}{s^2} \sum\_{i \in \mathcal{S}}\frac{1}{n} \sum_{j=1}^{n} \mathbb{E}[\|\mathbf{g}\_{j, t}^{(k)}-\nabla_{\mathbf{x}} f_j^{(k)}(\mathbf{x}\_{t}^{(k)}, \mathbf{y}\_{t}^{(k)}) \|^2] $
>
> $=\frac{1}{s}\frac{1}{n} \sum_{j=1}^{n} \mathbb{E}[\|\mathbf{g}\_{j, t}^{(k)}-\nabla_{\mathbf{x}} f_j^{(k)}(\mathbf{x}\_{t}^{(k)}, \mathbf{y}\_{t}^{(k)}) \|^2] $
>
> where the second step holds due to  $\mathbb{E}[\Big(\nabla\_{\mathbf{x}} f^{(k)}\_{i}(\mathbf{x}\_{t}^{(k)}, \mathbf{y}\_{t}^{(k)})-\mathbf{g}\_{i, t}^{(k)}+\frac{1}{n} \sum_{j=1}^{n} \mathbf{g}\_{j, t}^{(k)}- \frac{1}{n} \sum_{j=1}^{n} \nabla_{\mathbf{x}} f_j^{(k)}(\mathbf{x}\_{t}^{(k)}, \mathbf{y}\_{t}^{(k)}) \Big)] =0$, the third step holds due to $\mathbb{E}[||x-\mathbb{E}[x]||^2]\leq \mathbb{E}[||x||^2]$

---

> ### Comment · Action_Editor_UkXf · 2024-02-27
> **Thank you for your response**
>
> Dear Authors,
>
> Yes. You are right regarding this inequality. Sorry, I did not notice this since there are several missing steps here.
> I have another query regarding Eq (24). You also use the following inequality in the last step of Eq (24):
> $$
> E\big[|\frac{1}{s_{t+1}}\sum_{i}\big(\nabla f_i^k(x_{t+1}^k,y_{t+1}^k)-\nabla f_i^k(x_t^k,y_t^k)\big)|^2\big]\leq \frac{L^2}{s_{t+1}}E[|x_{t+1}^k-x_t^k|^2]+ \frac{L^2}{s_{t+1}}E[|y_{t+1}^k-y_t^k|^2].
> $$
> I also cannot follow this step. Note that the summand is not of expectation $0$ and therefore one cannot get a factor of $1/s_{t+1}$ on the right hand side (taking an average can decrease the variance by a factor of $s$ but cannot decrease the bias). That is, it seems that one can only get
> $$
> E\big[|\frac{1}{s_{t+1}}\sum_{i}\big(\nabla f_i^k(x_{t+1}^k,y_{t+1}^k)-\nabla f_i^k(x_t^k,y_t^k)\big)|^2\big]\leq L^2E[|x_{t+1}^k-x_t^k|^2]+L^2E[|y_{t+1}^k-y_t^k|^2].
> $$
> Would you please check it? Thanks.